# Latent Spherical Flow Policy for Reinforcement Learning with Combinatorial Actions

Lingkai Kong [* 1]  Anagha Satish [* 1]  Hezi Jiang [1]  Akseli Kangaslahti [1]  Andrew Ma [1]  Wenbo Chen [2]
Mingxiao Song [1]  Lily Xu [3]  Milind Tambe [1]

## Abstract

Reinforcement learning (RL) with combinatorial action spaces remains challenging because feasible action sets are exponentially large and governed by complex feasibility constraints, making direct policy parameterization impractical. Existing approaches embed task-specific value functions into constrained optimization programs or learn deterministic structured policies, sacrificing generality and policy expressiveness. We propose a solver-induced *latent spherical flow policy* that brings the expressiveness of modern generative policies to combinatorial RL while guaranteeing feasibility by design. Our method, LSFLOW, learns a *stochastic* policy in a compact continuous latent space via spherical flow matching, and delegates feasibility to a combinatorial optimization solver that maps each latent sample to a valid structured action. To improve efficiency, we train the value network directly in the latent space, avoiding repeated solver calls during policy optimization. To address the piecewise-constant and discontinuous value landscape induced by solver-based action selection, we introduce a smoothed Bellman operator that yields stable, well-defined learning targets. Empirically, our approach outperforms state-of-the-art baselines by an average of 20.6% across a range of challenging combinatorial RL tasks.

## 1. Introduction

Reinforcement learning (RL) has become a powerful framework for sequential decision-making. Recent work shows that RL can benefit from more expressive policy classes, such as diffusion (Sohl-Dickstein et al., 2015; Ho et al., 2020) and flow-based generative models (Lipman et al., 2023; Albergo & Vanden-Eijnden, 2023). These diffusion and flow policies (Yang et al., 2023; Psenka et al., 2024; Park et al., 2025; Zhang et al., 2025b) can represent rich, multimodal stochastic action distributions, enabling diverse exploratory behavior, and have demonstrated strong empirical performance in robot control benchmarks (e.g., locomotion and dexterous manipulation) (Wang et al., 2024; Ma et al., 2025a; Ren et al., 2025).

However, RL remains substantially challenged in *combinatorial action spaces*, where each action is a structured decision that must satisfy hard feasibility constraints, such as subset selection, routing, or scheduling. In these problems, the feasible set typically grows exponentially with problem size, making it intractable to explicitly represent a policy over all valid actions. Accordingly, prior work largely follows two directions: solver-based value optimization, which embeds a learned value network into a mixed-integer program (MIP) and extracts greedy actions by solving a constrained optimization problem over the feasible set (Xu et al., 2025); and optimization-induced decision layers, which learn structured policies end-to-end (Hoppe et al., 2025). The former often requires solver-compatible value architectures and can scale poorly with network complexity, while the latter yields deterministic policies, limiting expressiveness and reducing effective exploration in complex combinatorial landscapes.

A natural question is whether the expressiveness of diffusion and flow policies can be transferred to combinatorial RL. Direct transfer is nontrivial because these models are typically defined over continuous spaces and do not enforce discrete feasibility constraints or reflect the combinatorial structure of the feasible set. As a result, naive adaptation can produce infeasible actions, motivating approaches that couple generative expressiveness with feasibility guarantees.

In this work, we propose LSFLOW, a latent spherical flow policy for RL with combinatorial actions. Our key idea is to shift the burden of enforcing feasibility away from the policy network and onto a combinatorial optimization (CO)

---

[*]Equal contribution [1]Harvard University, Cambridge, MA, USA [2]Amazon. This work was conducted outside the author's role at Amazon. [3]Columbia University, New York, NY, USA. Correspondence to: Lingkai Kong <lingkaikong@g.harvard.edu>.

*Proceedings of the 43rd International Conference on Machine Learning*, Seoul, South Korea. PMLR 306, 2026. Copyright 2026 by the author(s).

solver. Specifically, we learn an expressive *stochastic* policy in a continuous latent space, where each latent sample specifies a linear objective for an optimization solver, then use a CO solver to convert the sample into a feasible structured action. This two-step parameterization combines the best of both worlds: the expressiveness of modern generative policies and feasibility guaranteed by the CO solver. Moreover, since a linear objective depends only on the objective *direction*, the latent space is naturally spherical, enabling us to learn the policy directly on the sphere via *spherical flow matching* (Chen & Lipman, 2024). Despite its compact form, the resulting policy remains fully expressive: with an appropriate spherical distribution, it can represent any stochastic policy over feasible combinatorial actions.

The expensive CO solver is the computational bottleneck. To make this approach efficient, we train the critic directly in the latent cost space to avoid calling the solver repeatedly during policy update. However, this introduces a central challenge: the mapping from latent cost to actions is discontinuous due to the solver, which can destabilize Bellman backups and value learning. We overcome this issue by introducing a *smoothed Bellman operator* that performs spherical smoothing using a von Mises–Fisher (vMF) kernel. We provide theoretical results showing that the resulting operator admits a unique fixed point and yields a well-defined smoothed value function that improves stability under solver-induced discontinuities.

Our main contributions are: (1) We propose a solver-augmented spherical flow policy that yields expressive stochastic policies over feasible structured actions. To the best of our knowledge, this is the first flow- or diffusion-based policy framework for RL with combinatorial action spaces. (2) We develop an efficient training method that learns the critic in the latent cost space and introduce a smoothed Bellman operator based on von Mises–Fisher (vMF) smoothing to mitigate solver-induced discontinuities and improve stability. (3) We demonstrate empirically that our method outperforms state-of-the-art baselines on public benchmarks and a real-world sexually transmitted infection (STI) testing application.

## 2. Background

### 2.1. Problem Definition

We study reinforcement learning in a discounted *combinatorial* Markov decision process (MDP) $(\mathcal{S}, \mathcal{A}, P, r, \gamma)$. At each timestep $h$, the agent observes a state $\boldsymbol{s}_h \in \mathcal{S}$ and selects a *structured* action $\boldsymbol{a}_h \in \mathcal{A}(\boldsymbol{s}_h)$, where the state-dependent feasible set $\mathcal{A}(\boldsymbol{s}) \subseteq \mathcal{A}$ is specified by combinatorial constraints (e.g., subset selection, matchings, or routes). Without loss of generality, we represent each feasible action as a binary decision vector $\boldsymbol{a} \in \{0,1\}^m$,

since any bounded integer decision can be encoded in binary (Dantzig, 1963). After executing $\boldsymbol{a}_h$, the environment returns a reward $r(\boldsymbol{s}_h, \boldsymbol{a}_h)$ and transitions to the next state $\boldsymbol{s}_{h+1} \sim P(\cdot \mid \boldsymbol{s}_h, \boldsymbol{a}_h)$.

Our goal is to learn a stochastic policy $\pi(\boldsymbol{a} \mid \boldsymbol{s})$ whose support is contained in $\mathcal{A}(\boldsymbol{s})$ for every $\boldsymbol{s} \in \mathcal{S}$, and that maximizes the expected discounted return

$$J(\pi) = \mathbb{E}_\pi \left[ \sum_{h=0}^{\infty} \gamma^h r(\boldsymbol{s}_h, \boldsymbol{a}_h) \right],$$

where the expectation is over trajectories induced by $\pi$ and the transition kernel $P$.

For any policy $\pi$, the action-value function is defined as

$$Q^\pi(\boldsymbol{s}, \boldsymbol{a}) \triangleq \mathbb{E}_\pi \left[ \sum_{t=0}^{\infty} \gamma^t r(\boldsymbol{s}_t, \boldsymbol{a}_t) \,\middle|\, \boldsymbol{s}_0 = \boldsymbol{s},\ \boldsymbol{a}_0 = \boldsymbol{a} \right].$$

Compared to standard RL, the exponentially large feasible set $\mathcal{A}(\boldsymbol{s})$ makes both greedy action selection from the action-value function and direct policy parameterization challenging.

### 2.2. Flow Matching

Flow matching (Albergo & Vanden-Eijnden, 2023; Lipman et al., 2023; Liu et al., 2023) is a continuous-time generative modeling framework based on ordinary differential equations (ODEs). In contrast to denoising diffusion models, which are typically formulated via stochastic differential equations (SDEs), flow models often admit simpler training and faster sampling while achieving competitive or improved generation quality (Esser et al., 2024; Lipman et al., 2024). Concretely, a flow model transports samples from a base distribution $p_0$ to a target distribution $p_1$ by integrating

$$\frac{d\boldsymbol{x}_t}{dt} = v_\theta(\boldsymbol{x}_t, t), \qquad t \in [0, 1], \tag{1}$$

where $v_\theta(\cdot, t)$ is a time-dependent velocity field parameterized by $\theta$. Starting from initial sample $\boldsymbol{x}_0 \sim p_0$, integrating Eq. (1) to $t = 1$ produces $\boldsymbol{x}_1$ whose distribution approximates $p_1$.

To learn $v_\theta$, flow matching regresses the model velocity to a target path derivative induced by a chosen interpolation between $p_0$ and $p_1$. Specifically, sample $\boldsymbol{x}_0 \sim p_0$, $\boldsymbol{x}_1 \sim p_1$ independently, and $t \sim \text{Unif}[0, 1]$. Let $\boldsymbol{x}_t = \phi_t(\boldsymbol{x}_0, \boldsymbol{x}_1)$ be an interpolation and define the target velocity $u_t(\boldsymbol{x}_t, t) \triangleq \frac{d}{dt}\phi_t(\boldsymbol{x}_0, \boldsymbol{x}_1)$. The general flow-matching objective is

$$\min_\theta \ \mathbb{E}_{\boldsymbol{x}_0 \sim p_0,\, \boldsymbol{x}_1 \sim p_1,\, t \sim \text{Unif}[0,1]} \left[ \left\| v_\theta(\boldsymbol{x}_t, t) - u_t(\boldsymbol{x}_t, t) \right\|^2 \right].$$

The most common choice of interpolation is the linear path (Liu et al., 2023):

$$\boldsymbol{x}_t = (1 - t)\boldsymbol{x}_0 + t\boldsymbol{x}_1, \qquad u_t(\boldsymbol{x}_t, t) = \boldsymbol{x}_1 - \boldsymbol{x}_0,$$

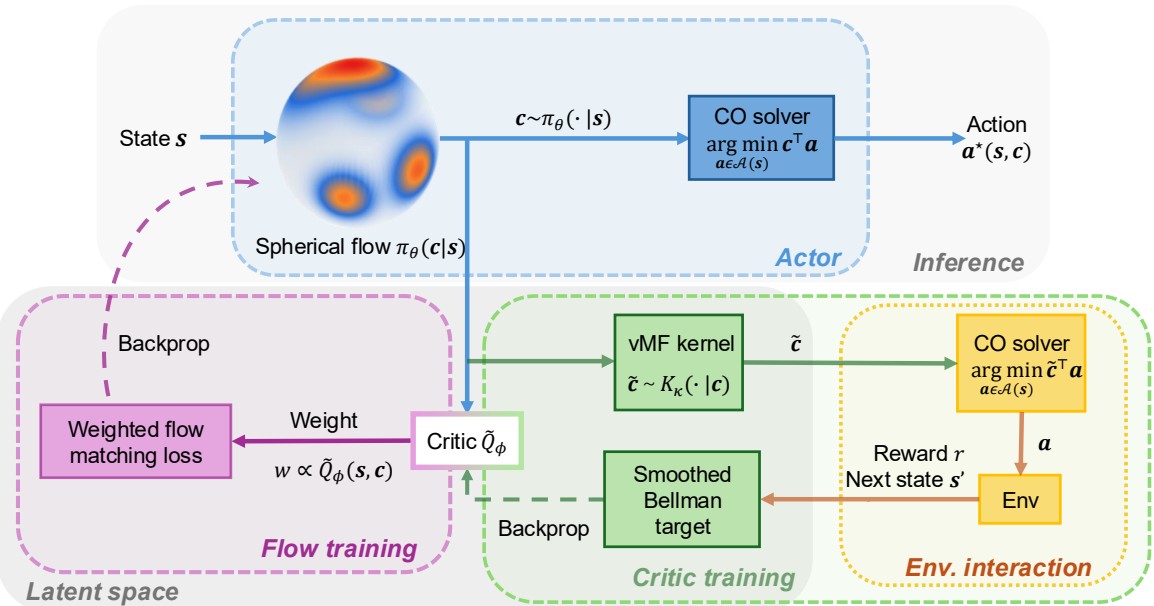

*Figure 1.* **Overall framework of LSFLOW.** To enable expressive *stochastic* policies under hard combinatorial constraints, we shift policy learning to a continuous spherical *cost-direction* space and let a combinatorial optimization (CO) solver enforce feasibility (Section 3.1). Crucially, both policy and critic are learned directly in this latent cost space. We train the flow policy $\pi_\theta$ via weighted flow matching, using weights $w \propto \tilde{Q}_\phi(s, c)$ predicted by a latent-space critic (Section 3.2). To stabilize cost-space critic learning under solver-induced discontinuities, we apply vMF smoothing by sampling $\tilde{c} \sim K_\kappa(\cdot \mid c)$ when constructing Bellman targets (Section 3.3).

which yields a closed-form target velocity.

## 2.3. Flow Policies

In continuous-action RL, flow models naturally induce expressive stochastic policies by conditioning the velocity field on the state, i.e., $v_\theta(\cdot, t; s)$ (Park et al., 2025; Zhang et al., 2025b; McAllister et al., 2025). A policy sample is obtained by drawing $x_0 \sim p_0$, integrating the state-conditional ODE, and outputting $a \triangleq x_1$, which defines a distribution $\pi_\theta(a \mid s)$ without explicitly enumerating actions.

In combinatorial decision making, however, actions must lie in a discrete feasible set $\mathcal{A}(s)$. Therefore, a valid policy must place probability mass only within $\mathcal{A}(s)$, rather than on an unconstrained subset of $\mathbb{R}^m$. Feasibility preservation is a central challenge that we address next.

## 3. Proposed Method

Here, we present our method, LSFLOW. We begin with an expressive policy representation that combines flow matching with a combinatorial optimization solver, yielding a flexible distribution over *feasible* actions without enumerating the combinatorial action space. We then introduce an efficient objective for policy optimization based on a smoothed Bellman operator, and provide practical implementation details.

### 3.1. Combinatorial Flow Policy Representation

We address the challenge of representing stochastic policies over constrained combinatorial action spaces by introducing a *solver-induced policy representation* that decouples stochasticity from feasibility. The key idea is a two-stage construction: (i) we learn a continuous, state-conditional distribution over *cost vectors* using flow matching, and (ii) we map each sampled cost vector to a feasible combinatorial action via a downstream optimization solver. This induces an expressive distribution over feasible actions while avoiding explicit enumeration of $\mathcal{A}(s)$.

**Solver-induced flow policy.** Let $c \in \mathbb{R}^m$ denote a *cost vector* that parameterizes a linear objective for a combinatorial solver over the feasible action set $\mathcal{A}(s)$. We define the solver mapping

$$a^\star(s, c) = \arg\min_{a \in \mathcal{A}(s)} \ c^\top a \,, \qquad (2)$$

where $\mathcal{A}(s)$ encodes the combinatorial feasibility constraints. Given state $s$, the policy first samples $c \sim \pi_\theta(\cdot \mid s)$ then executes the solver output $a = a^\star(s, c)$. This construction defines a stochastic policy over $\mathcal{A}(s)$ as the pushforward of $\pi_\theta(\cdot \mid s)$ through $a^\star(\cdot)$. Feasibility is guaranteed by the solver, while stochasticity is captured entirely by the learned flow distribution over cost vectors.

*Example (routing).* In a routing problem, $a \in \{0, 1\}^m$ indicates selected edges and $\mathcal{A}(s)$ enforces connectivity, flow,

and budget constraints that depend on the current state $s$ (e.g., available edges, travel times, or blocked roads). Each entry of $c$ specifies a state-dependent edge cost, and $a^\star(s, c)$ returns the minimum-cost feasible route under these costs.

**Lemma 3.1** (Positive scale invariance of the solver mapping). *Fix any state $s$ with $\mathcal{A}(s) \neq \emptyset$, and let $a^\star(s, c) = \arg\min_{a \in \mathcal{A}(s)} c^\top a$. Then the solver mapping is positively scale invariant: for any $\alpha > 0$,*

$$a^\star(s, \alpha c) = a^\star(s, c).$$

By Lemma 3.1, for any fixed state $s$, the solver output $a^\star(s, c)$ depends only on the *direction* of $c$, not its magnitude: any positive rescaling leaves the induced action unchanged. To remove this redundancy and obtain a compact policy domain, we restrict cost vectors to the unit sphere,

$$\mathcal{C} = \mathbb{S}^{m-1} := \{c \in \mathbb{R}^m : \|c\|_2 = 1\}.$$

Accordingly, we model $\pi_\theta(c \mid s)$ as a distribution on $\mathbb{S}^{m-1}$. This spherical parameterization matches the solver's invariance and provides a well-behaved manifold on which we apply flow matching to learn rich, state-conditional cost distributions.

Although each cost vector induces only a linear objective for the solver, this does not limit the policies it can express: we now show that, with a suitable distribution over cost directions on the sphere, the solver-induced construction can represent *any* stochastic policy over $\mathcal{A}(s)$.

**Proposition 3.2** (Exact expressivity of solver-induced policies). *Fix a state $s$ and assume the feasible action set is finite. Then for any target distribution $\mu$ over $\mathcal{A}(s)$, there exists a distribution $\pi$ over $\mathbb{S}^{m-1}$ such that if we sample $c \sim \pi$ and take the solver action $a = a^\star(s, c)$, then $a$ is distributed according to $\mu$. Equivalently, for all $a \in \mathcal{A}(s)$,*

$$\mathbb{P}[a^\star(s, c) = a] = \mu(a).$$

*Here, $\mathbb{P}[\cdot]$ denotes probability with respect to the randomness of drawing $c \sim \pi$.*

**Spherical flow policy.** To parameterize a stochastic policy over unit-norm cost vectors $c \in \mathbb{S}^{m-1}$, we model $\pi_\theta(c \mid s)$ using *spherical flow matching* (Chen & Lipman, 2024). Concretely, we learn a time-dependent vector field $v_\theta(c, s, t)$ and generate samples $c_1 \sim \pi_\theta(\cdot \mid s)$ by integrating a projected ODE that remains on $\mathbb{S}^{m-1}$:

$$\frac{dc_t}{dt} = \Pi_{c_t} v_\theta(c_t, s, t), \qquad c_0 \sim p_0, \quad c_t \in \mathbb{S}^{m-1}, \quad (3)$$

where $p_0$ is a fixed base distribution on $\mathbb{S}^{m-1}$ (e.g., the uniform distribution), and $\Pi_c = \mathbf{I} - cc^\top$ is the orthogonal projection onto the tangent space at $c$. This projection ensures that the flow stays on the spherical manifold while retaining the expressive capacity of continuous-time generative models.

*Remark* 3.3. The reason we choose flow matching over diffusion is twofold: (1) spherical flow matching achieves substantially better performance than spherical diffusion models, as shown in Chen & Lipman (2024); (2) implementing spherical diffusion models is considerably more complex in practice.

### 3.2. Efficient Policy Learning in Latent Cost Space

Our goal is to optimize a solver-induced spherical flow policy to maximize expected return. Recall that the policy outputs a cost direction $c$ on the sphere, and the environment action is obtained only *afterwards* via the CO solver, $a = a^\star(s, c)$. End-to-end differentiation through both the flow sampler and the solver is computationally expensive, and the solver mapping is non-smooth. Instead, we perform policy improvement *entirely in latent space* via a *proposal-and-weight* update. Crucially, we compute reweighting scores using a *cost-space critic* $\widetilde{Q}(s, c)$, which avoids repeated solver calls during the policy optimization loop.

**Weighted spherical flow-matching objective.** At iteration $k$, we sample proposal cost directions $c_1 \sim \pi_k(\cdot \mid s)$ and improve the next policy by refitting the spherical flow model to a *weighted* version of these latent samples. Each proposal is assigned an importance weight

$$w(s, c) \propto \exp\left(\frac{1}{\lambda} Q(s, a^\star(s, c))\right),$$

where $\lambda > 0$ controls the update aggressiveness (smaller $\lambda$ yields stronger reweighting). We then minimize the weighted spherical flow-matching loss

$$\mathcal{L}(\theta) = \mathbb{E}\Big[w(s, c_1) \big\|\Pi_{c_t} v_\theta(c_t, s, t) - u(c_t, s, t)\big\|_2^2\Big], \quad (4)$$

where the expectation is over $s \sim \mathcal{D}$, $c_1 \sim \pi_k(\cdot \mid s)$, $c_0 \sim p_0$, and $t \sim \mathcal{U}(0, 1)$. Here $\mathcal{D}$ is the replay buffer, $c_t$ is the geodesic interpolation between $c_0$ and $c_1$, $\Pi_{c_t}$ projects onto the tangent space at $c_t$, and $u(c_t, s, t)$ is the spherical flow-matching target (Appendix D). Larger $w(s, c_1)$ upweights high-value *latent directions*, shifting $\pi_{k+1}$ toward regions of the cost sphere that produce better solver outputs.

**Connection to KL-regularized policy improvement.** Prior work on diffusion policies shows that exponential reweighting corresponds to a KL-regularized policy improvement step (Ma et al., 2025a); a related reweighting also appears in offline energy-weighted flow matching (Zhang et al., 2025a). We extend this interpretation to online combinatorial RL in the latent cost space. In particular, the updated cost policy $\pi_{k+1}$ solves the KL-regularized update (Tomar et al., 2022)

$$\pi_{k+1}(\cdot \mid s) \in \arg\max_{\pi(\cdot \mid s)} \mathbb{E}_{c \sim \pi(\cdot \mid s)}[Q(s, a^\star(s, c))]$$
$$- \lambda \, \mathrm{KL}\big(\pi(\cdot \mid s) \,\|\, \pi_k(\cdot \mid s)\big).$$

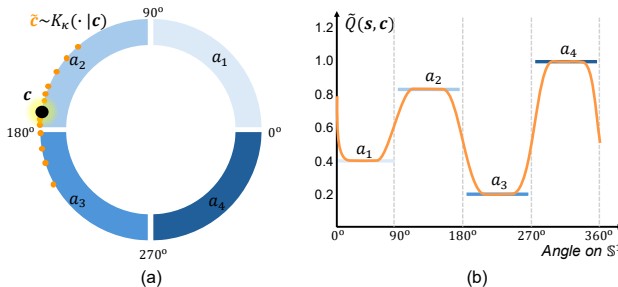

*Figure 2.* (a) The solver partitions the latent space on $\mathbb{S}^1$ into regions that map to different actions. We smooth this partition by averaging with a vMF kernel $K_\kappa(\cdot|\boldsymbol{c})$. (b) The corresponding value $\tilde{Q}(\boldsymbol{s}, \boldsymbol{c})$ is originally piecewise constant across regions, but becomes a smooth function after vMF smoothing.

The KL term acts as a trust region—analogous to PPO (Schulman et al., 2017) and TRPO (Schulman et al., 2015)—and stabilizes learning. A proof is provided in Section I.

**Computing weights efficiently via a cost-space critic.** A naive implementation would learn an action-value function $Q(\boldsymbol{s}, \boldsymbol{a})$ and score each proposal by evaluating $Q(\boldsymbol{s}, \boldsymbol{a}^\star(\boldsymbol{s}, \boldsymbol{c}))$, which would require calling the solver *inside the policy update loop* and become a major bottleneck when many latent proposals are used per state. To avoid this, we learn a critic *directly on the latent cost space*:

$$\widetilde{Q}_\phi(\boldsymbol{s}, \boldsymbol{c}) := Q(\boldsymbol{s}, \boldsymbol{a}^\star(\boldsymbol{s}, \boldsymbol{c})). \quad (5)$$

In practice, $\widetilde{Q}_\phi(\boldsymbol{s}, \boldsymbol{c})$ is a neural network trained from rewards generated by executing $\boldsymbol{a}^\star(\boldsymbol{s}, \boldsymbol{c})$ in the environment (see Section 3.4). This amortizes solver cost: the solver is invoked once per environment step to execute an action, rather than repeatedly during policy optimization. As a result, the update in Eq. (4) depends only on $(\boldsymbol{s}, \boldsymbol{c})$ pairs and can be performed without additional solver calls.

### 3.3. On-Sphere Smoothing

The cost-space critic in Eq. (5) amortizes solver calls and enables efficient policy optimization. However, it also introduces a stability challenge: the mapping $\boldsymbol{c} \mapsto \boldsymbol{a}^\star(\boldsymbol{s}, \boldsymbol{c})$ is *piecewise constant* over the sphere. Intuitively, the solver returns the same action throughout each optimality region and switches abruptly at region boundaries (e.g., as characterized by the normal fan of a polyhedral feasible set). As a result, the induced cost-space critic $\widetilde{Q}(\boldsymbol{s}, \boldsymbol{c}) = Q(\boldsymbol{s}, \boldsymbol{a}^\star(\boldsymbol{s}, \boldsymbol{c}))$ can be highly non-smooth in $\boldsymbol{c}$: even a tiny perturbation of $\boldsymbol{c}$ may cross a boundary and cause a discontinuous jump in $\boldsymbol{a}^\star(\boldsymbol{s}, \boldsymbol{c})$. This non-smoothness leads to unstable bootstrap targets and high-variance gradients when fitting $\widetilde{Q}_\phi$ and updating the policy.

To stabilize learning, we introduce *kernel smoothing on the sphere*. The key idea is to replace the dependence on a single cost direction by a local average over nearby directions, which removes the discontinuities *in the learning target* induced by the solver.

**vMF smoothing kernel.** Fix a concentration parameter $\kappa > 0$. The von Mises–Fisher (vMF) kernel (Núñez-Antonio & Gutiérrez-Peña, 2005) defines a distribution over directions $\tilde{\boldsymbol{c}}$ in a neighborhood of a center direction $\boldsymbol{c}$ on the sphere:

$$K_\kappa(\tilde{\boldsymbol{c}} \mid \boldsymbol{c}) \propto \exp(\kappa\, \boldsymbol{c}^\top \tilde{\boldsymbol{c}}), \qquad \boldsymbol{c}, \tilde{\boldsymbol{c}} \in \mathbb{S}^{m-1}. \quad (6)$$

Larger $\kappa$ yields a more concentrated kernel, with $K_\kappa(\cdot \mid \boldsymbol{c})$ peaking more sharply around $\boldsymbol{c}$.

**Smoothed Bellman operator.** Given a policy $\pi(\boldsymbol{c} \mid \boldsymbol{s})$, we define a smoothed Bellman operator that smooths (i) the current-step cost direction before solving, and (ii) the bootstrap direction before evaluating the critic. For any bounded $Q : \mathcal{S} \times \mathbb{S}^{m-1} \to \mathbb{R}$,

$$(\mathcal{T}_\kappa^\pi Q)(\boldsymbol{s}, \boldsymbol{c}) = \mathbb{E}[r(\boldsymbol{s}, \boldsymbol{a}^\star(\boldsymbol{s}, \tilde{\boldsymbol{c}})) + \gamma\, Q(\boldsymbol{s}', \tilde{\boldsymbol{c}}')], \quad (7)$$

where the expectation is over $\tilde{\boldsymbol{c}} \sim K_\kappa(\cdot \mid \boldsymbol{c})$, $\boldsymbol{s}' \sim P(\cdot \mid \boldsymbol{s}, \boldsymbol{a}^\star(\boldsymbol{s}, \tilde{\boldsymbol{c}}))$, $\boldsymbol{c}' \sim \pi(\cdot \mid \boldsymbol{s}')$, and $\tilde{\boldsymbol{c}}' \sim K_\kappa(\cdot \mid \boldsymbol{c}')$.

Compared to the standard Bellman operator, $\mathcal{T}_\kappa^\pi$ replaces the discontinuous dependence on a single solver output $\boldsymbol{a}^\star(\boldsymbol{s}, \boldsymbol{c})$ with a local average over $\boldsymbol{a}^\star(\boldsymbol{s}, \tilde{\boldsymbol{c}})$ for nearby directions $\tilde{\boldsymbol{c}}$. It likewise smooths the bootstrap argument by evaluating $Q$ at a perturbed next-step direction $\tilde{\boldsymbol{c}}'$. As a result, the learning targets vary smoothly with $\boldsymbol{c}$, improving stability and reducing gradient variance, while still enforcing feasibility through the solver $\boldsymbol{a}^\star(\boldsymbol{s}, \cdot)$.

**Assumption 3.4** (Boundedness and measurability). *Rewards $r$ are bounded, transitions $P$ and policy $\pi$ are measurable, and $Q$ is bounded.*

**Theorem 3.5** (Contraction and smoothness of the fixed point). *Let $\mathcal{B}_\infty$ be the space of bounded functions $Q : \mathcal{S} \times \mathbb{S}^{m-1} \to \mathbb{R}$ equipped with the sup norm $\|Q\|_\infty := \sup_{\boldsymbol{s} \in \mathcal{S},\, \boldsymbol{c} \in \mathbb{S}^{m-1}} |Q(\boldsymbol{s}, \boldsymbol{c})|$. For a fixed $\boldsymbol{s}$, the notation $Q(\boldsymbol{s}, \cdot) \in C^\infty(\mathbb{S}^{m-1})$ means that the map $\boldsymbol{c} \mapsto Q(\boldsymbol{s}, \boldsymbol{c})$ is infinitely differentiable on $\mathbb{S}^{m-1}$. Under Assumption 3.4 with $0 \le \gamma < 1$, and using the vMF kernel $K_\kappa$ in Eq. (6), the following hold:*

1. *($\gamma$-contraction) $\mathcal{T}_\kappa^\pi$ is a $\gamma$-contraction on $(\mathcal{B}_\infty, \|\cdot\|_\infty)$ and admits a unique fixed point $Q_\kappa^\pi$.*

2. *($C^\infty$ smoothing in $\boldsymbol{c}$) For any $Q \in \mathcal{B}_\infty$ and any $\boldsymbol{s}$, $(\mathcal{T}_\kappa^\pi Q)(\boldsymbol{s}, \cdot) \in C^\infty(\mathbb{S}^{m-1})$.*

3. *(Regularity of the fixed point) Consequently, for every $\boldsymbol{s}$, $Q_\kappa^\pi(\boldsymbol{s}, \cdot) \in C^\infty(\mathbb{S}^{m-1})$.*

Theorem 3.5 establishes that the smoothed Bellman operator admits a unique, globally well-defined fixed point whose value function is infinitely differentiable in the cost direction. This smoothness is what makes the operator useful in practice: it yields stable, low-variance bootstrap targets that vary smoothly with $c$, which is precisely what stabilizes critic and policy updates under the otherwise discontinuous solver mapping.

This stability, however, comes from modifying the operator itself: $\mathcal{T}_\kappa^\pi$ no longer shares the fixed point of the original Bellman operator $\mathcal{T}^\pi$. At any finite $\kappa$ this introduces a bias—a deliberate trade-off against the variance reduction above, which we study empirically in Section 5.3. A natural question is therefore whether this bias is benign or whether smoothing fundamentally distorts the quantity we ultimately care about. The following result shows the former: the smoothed fixed point is *consistent*, recovering the unsmoothed value as the kernel concentrates.

**Theorem 3.6** (Consistency of the smoothed fixed point). *For each reachable state $s$, let $\mathcal{B}_s := \{c \in \mathbb{S}^{m-1} : \arg\min_{a \in \mathcal{A}(s)} c^\top a$ is not a singleton$\}$ denote the solver-switching boundary, and assume $\pi(\cdot \mid s)$ assigns zero mass to $\mathcal{B}_s$. Then for every reachable $s$ and every $c \notin \mathcal{B}_s$,*

$$Q_\kappa^\pi(s, c) \longrightarrow Q^\pi(s, c) \qquad as\ \kappa \to \infty,$$

*where $Q^\pi$ is the fixed point of the unsmoothed Bellman operator $\mathcal{T}^\pi$. Since $\mathcal{B}_s$ has surface measure zero, this convergence also holds for surface-a.e. $c \in \mathbb{S}^{m-1}$, and $Q_\kappa^\pi(s, C) \to Q^\pi(s, C)$ almost surely when $C \sim \pi(\cdot \mid s)$.*

*Remark* 3.7. A global sup-norm bound $\|Q_\kappa^\pi - Q^\pi\|_\infty \to 0$ is not achievable in general: by Theorem 3.5, $Q_\kappa^\pi(s, \cdot) \in C^\infty(\mathbb{S}^{m-1})$ for every $\kappa > 0$, whereas $Q^\pi(s, \cdot)$ is typically piecewise constant with jumps across $\mathcal{B}_s$, so a uniform limit cannot recover $Q^\pi$; almost-everywhere convergence is therefore the appropriate notion here. The boundary-mass assumption in Theorem 3.6 is mild: it holds whenever $\pi(\cdot \mid s)$ admits a density with respect to surface measure on $\mathbb{S}^{m-1}$, which is the case for the spherical flow policy used throughout this paper. The convergence can be made quantitative under additional regularity of $\pi$ near $\mathcal{B}_s$; we leave a precise rate to future work.

### 3.4. Practical Implementation

We now summarize the training loop that instantiates the flow-policy update and the smoothed Bellman operator from Section 3.3; full pseudocode is provided in Algorithm 1.

**Data collection.** At each training step, the policy samples a cost direction $c \sim \pi(\cdot \mid s)$ and applies an on-sphere perturbation $\tilde{c} \sim K_\kappa(\cdot \mid c)$ before solving. We execute $a = a^\star(s, \tilde{c})$ in the environment and store the *center* direction together with the transition, i.e., $(s, c, r, s')$. This matches the smoothing in Eq. (7).

**Critic update.** Given a replayed transition $(s, c, r, s')$, we sample $c' \sim \pi_\theta(\cdot \mid s')$ and draw $\tilde{c}'^{(j)} \sim K_\kappa(\cdot \mid c')$ for $J$ perturbed directions, then form the smoothed target

$$y_\kappa \leftarrow r + \gamma \frac{1}{J} \sum_{j=1}^J \widetilde{Q}_{\bar\phi}(s', \tilde{c}'^{(j)}) . \qquad (8)$$

We fit the critic by minimizing the squared Bellman error $\min_\phi \mathbb{E}\left[\left(\widetilde{Q}_\phi(s, c) - y_\kappa\right)^2\right]$. By construction, Eq. (8) is a Monte Carlo estimate of the smoothed backup in Eq. (7), yielding targets that vary smoothly with $c$ and mitigating solver-induced discontinuities.

**Policy update.** We update the spherical flow policy using the reweighted flow-matching objective in Eq. (4), which shifts probability mass toward cost directions that induce higher-value solver outputs.

## 4. Related Works

**Diffusion/flow policies for RL.** Diffusion (Sohl-Dickstein et al., 2015; Ho et al., 2020) and flow-based generative models (Lipman et al., 2023; Albergo & Vanden-Eijnden, 2023) enable *expressive stochastic policies* that capture multi-modal action distributions beyond Gaussian actors. Early work focused on *offline RL*, using conservative, behavior-regularized training to stay within the data support, as in Diffusion Q-learning (Wang et al., 2023) and sampling-efficient variants (Kang et al., 2023). Recent extensions (Choi et al., 2026) bring diffusion policies to *online* RL via environment interaction, including action-gradient methods (Yang et al., 2023; Psenka et al., 2024; Li et al., 2024), value-/advantage-weighted updates (Ding et al., 2024; Ma et al., 2025a), proximity-based regularization (Ding et al., 2025), and backpropagation through multi-step generative rollouts (Wang et al., 2024; Celik et al., 2025). In parallel, *flow-matching* policies offer faster sampling and have been linked to RL objectives in both offline (Zhang et al., 2025a; Park et al., 2025) and online regimes (Zhang et al., 2025b; McAllister et al., 2025). These approaches work well in continuous control, but they do not enforce discrete feasibility constraints and thus do not directly apply to structured combinatorial action spaces.

Note that concurrent work (Ma et al., 2025b) has begun exploring *discrete* diffusion policies for RL with discrete action spaces. However, this method targets standard discrete actions rather than *combinatorial* actions, and therefore does not directly enforce hard feasibility constraints.

**RL with combinatorial actions.** For large discrete action spaces, prior work uses action embeddings and approximate maximization to reduce action-selection cost (Dulac-Arnold et al., 2015), but it becomes impractical when the feasible set is exponential and does not enforce hard constraints.

| Method | Reward (↑) | | | | Avg. (↑) | Avg. Time (h) (↓) |
|---|---|---|---|---|---|---|
| | **Dyn. Sched.** | **Dyn. Routing** | **Dyn. Assign.** | **Dyn. Interv.** | | |
| Random | $10.12{\pm}0.84$ | $11.11{\pm}0.36$ | $13.28{\pm}0.80$ | $8.91{\pm}0.08$ | 10.85 | – |
| Greedy | $15.12{\pm}1.56$ | $11.58{\pm}0.46$ | $20.15{\pm}1.00$ | $10.55{\pm}0.67$ | 14.35 | – |
| DQN-Sampling (He et al., 2016) | $16.89{\pm}0.95$ | $18.29{\pm}1.49$ | $22.25{\pm}2.27$ | $15.16{\pm}0.25$ | 18.15 | 0.52 |
| SEQUOIA (Xu et al., 2025) | $24.00{\pm}1.44$ | $20.99{\pm}1.92$ | $32.99{\pm}3.45$ | $10.84{\pm}0.89$ | 22.21 | 9.11 |
| SRL (Hoppe et al., 2025) | $24.50{\pm}1.09$ | $25.32{\pm}1.49$ | $28.46{\pm}0.82$ | $13.36{\pm}1.93$ | 22.91 | 3.89 |
| **Ours** | $\mathbf{28.85}{\pm}1.48$ | $\mathbf{28.51}{\pm}2.60$ | $\mathbf{35.93}{\pm}2.71$ | $\mathbf{17.21}{\pm}0.39$ | **27.62** | 1.29 |

*Table 1.* Rewards across tasks (higher is better). Average training time is reported in hours (lower is better).

In combinatorial domains, He et al. (2016) instead sample a fixed set of feasible candidate actions and use the critic to pick the best as a proxy for greedy maximization. Related theory provides tractable guarantees under restricted function classes (e.g., linear approximation) (Tkachuk et al., 2023), which may not extend to expressive nonlinear critics.

More recent methods couple RL with combinatorial optimization solvers to enforce feasibility by design. A prominent line performs *solver-based value optimization*, selecting actions by solving a mixed-integer program (MIP) that maximizes a learned value over the feasible set: Delarue et al. (2020) formulate greedy improvement as a MIP augmented with a learned value-to-go term, but focus on vehicle routing in deterministic MDPs; Xu et al. (2025) propose SEQUOIA, which embeds a Q-network into a MIP to compute greedy actions in stochastic sequential settings such as coupled restless bandits. While effective, these approaches typically require the value function to be solver-embeddable, which can restrict the critic class. Another line integrates optimization into the *policy parameterization* itself: Hoppe et al. (2025) propose Structured Reinforcement Learning (SRL), an actor–critic framework that trains a solver-induced actor end-to-end via stochastic perturbations and Fenchel–Young losses (Blondel et al., 2020). However, SRL yields deterministic structured policies, which can limit expressiveness and exploration in complex combinatorial landscapes. Overall, existing work highlights a central trade-off among feasibility guarantees, generalizability, and policy expressiveness in RL with combinatorial actions.

# 5. Experiments

We evaluate LSFLOW on a public benchmark and on a real-world sexually transmitted infection testing task.[1] We also conduct ablation studies to validate key design choices.

[1]Our code is available at `https://github.com/anagha-satish/combinatorial-diffusion`

## 5.1. Public Benchmarks

We first evaluate our method on the public benchmark suite of Xu et al. (2025), which defines four sequential decision-making tasks with combinatorially constrained actions. In each task, the learner repeatedly selects a feasible structured action, observes reward feedback, and optimizes long-horizon return under stochastic dynamics: (1) *Dynamic scheduling* repeatedly selects a limited set of service requests and assigns them to feasible time windows at each step, where completing (or delaying) service changes which requests remain and how rewards evolve over time; (2) *Dynamic routing* plans a bounded-length route on a graph (e.g., delivery), where the visited nodes determine both immediate reward and future states; (3) *Dynamic assignment* repeatedly matches limited-capacity resources to incoming demands over time, where today's allocation affects future availability and rewards; and (4) *Dynamic intervention* selects a constrained subset of interventions to apply to a population at each step where intervening on an individual changes their future state and thus the rewards available in later rounds. Additional environment details are provided in Section J.1.

**Experimental setup.** We compare against two simple heuristics, RANDOM and GREEDY; a sampling-based baseline (He et al., 2016) that draws 100 feasible actions and selects the one with the highest critic value; and two state-of-the-art combinatorial RL methods: SEQUOIA (Xu et al., 2025), which embeds a learned Q-network into a mixed-integer program (MIP) for greedy action selection, and SRL (Hoppe et al., 2025), which learns a deterministic structured policy. For a fair comparison, all RL methods use the same critic-network architecture and the same number of warm-up steps, and are then trained until convergence. At test time, we report average return over 50 evaluation episodes. Additional baseline details and hyperparameters are provided in Section J.1.

**Experimental results.** Table 1 summarizes the performance of all methods. LSFLOW achieves the highest reward on all four tasks, improving over the strongest baseline SRL by 20.6% on average. We attribute these gains to the

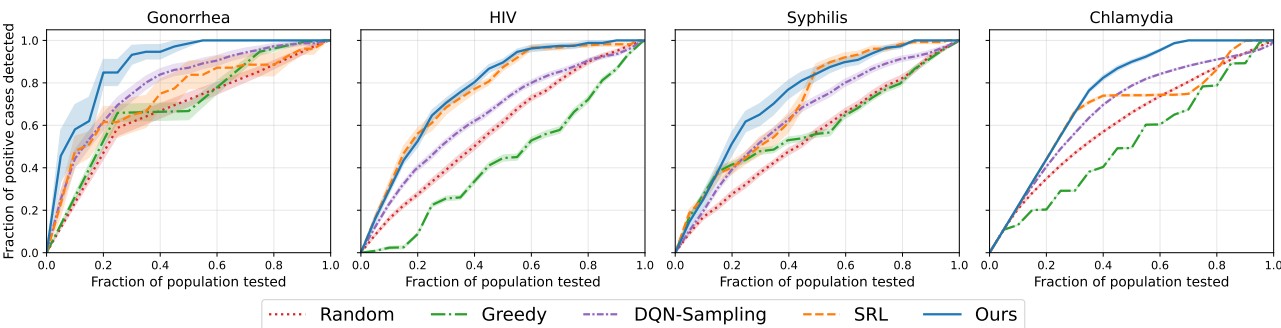

*Figure 3.* Performance on sexually transmitted infection testing

expressiveness of our *stochastic* policy class. SRL learns a deterministic structured policy, which can limit representational capacity and reduce exploration, potentially leading to suboptimal local optima. In contrast, our approach uses flow matching to learn a rich state-conditional distribution in a continuous latent space, and the downstream solver maps each latent sample to a feasible structured action. This combination yields an expressive stochastic policy over the constrained combinatorial action set, effectively transferring the benefits of diffusion/flow policies to combinatorial RL.

In addition to stronger performance, LSFLOW is also computationally efficient. Our approach is approximately $3.0\times$ faster than SRL in average training time. This speedup arises because SRL differentiates through the combinatorial solver by sampling multiple perturbations of each cost vector, which requires solving a combinatorial optimization problem for every perturbed instance during training. In contrast, our method avoids differentiating through the solver: we train the flow policy in the latent cost space with a reweighted objective and learn the critic directly in the latent space, so each environment step requires only a single solver call. SEQUOIA is even slower, as it solves an embedded mixed-integer program whose number of decision variables grows with the network size, and it typically relies on ReLU-based MLP architectures. These design choices can be restrictive and may not scale well to other problem settings, as we show next.

### 5.2. Sexually Transmitted Infection Testing

We next evaluate LSFLOW on a real-world *sexually transmitted infection (STI)* testing task. STI testing is a core public-health intervention, and the WHO recommends network-based strategies that leverage contact networks to reach high-risk individuals (World Health Organization, 2021). Specifically, we model the population as a contact network, where nodes represent individuals and edges denote reported sexual interactions, and focus on screening for gonorrhea, HIV, syphilis, and chlamydia. At each step, a health worker selects individuals to test. Testing a node reveals its in-

fection status and yields reward when a positive case is detected. Actions are further constrained by a *frontier* rule: tests can only be assigned to individuals adjacent to at least one previously tested node. While network-based testing is recommended, few works provide concrete sequential decision rules; for example, Choo et al. (2025); Kangaslahti et al. (2026) study a simplified setting that selects a single node at a time. In contrast, we study a batch-testing variant that better reflects practice: the agent selects a budgeted *set* of nodes at each step, inducing a constrained combinatorial action space. This setting admits no known optimal solution method, making it a natural testbed for evaluating RL-based approaches. Additional environment details are provided in Section K.1.

**Experimental setup.** We construct sexual interaction graphs from the de-identified public-use dataset released by ICPSR (Morris & Rothenberg, 2011). Since this domain is graph-structured, we use a Graph Isomorphism Network (GIN) backbone (Xu et al., 2019) for the critic in all RL-based methods. We exclude SEQUOIA from this experiment because it is restricted to MLP-based value networks, whereas this task requires a graph neural network for effective representation learning. Additional baseline details and hyperparameters are provided in Section K.2.

**Experimental results.** We evaluate performance by the fraction of positive cases detected versus the fraction of the population tested. Figure 3 reports results on four real disease networks (gonorrhea, HIV, syphilis, and chlamydia). Overall, our method achieves higher detection efficiency: for a fixed testing budget, it identifies a larger fraction of positive cases, with the largest gains in the low-to-mid budget regime where targeted testing matters most. In particular, LSFLOW consistently improves early-stage detection over both RANDOM and GREEDY across all diseases. We outperform SRL on chlamydia and gonorrhea and match it on HIV; on syphilis, we achieve stronger *early* detection—often the most operationally valuable regime—while SRL becomes comparable only at larger testing fractions. These results

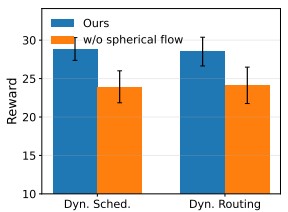 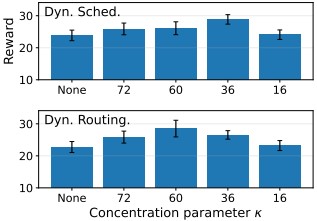

*Figure 4.* Ablation study. **Left:** effect of training the flow directly on the sphere. **Right:** effect of vMF smoothing. Here $\kappa$ is the concentration parameter: larger $\kappa$ yields weaker smoothing, while smaller $\kappa$ yields stronger smoothing. `None` denotes no smoothing.

suggest that an expressive stochastic policy is especially effective for navigating the frontier constraint and prioritizing high-yield test candidates under limited testing budgets.

### 5.3. Ablation Study

**Spherical flow.** We ablate learning the cost-direction distribution *on the sphere* versus in unconstrained Euclidean space. Specifically, we replace spherical flow matching with an Euclidean flow model (Lipman et al., 2023) that outputs cost vectors in $\mathbb{R}^m$ (same architecture and weighted objective) and map them to actions with the same solver. The only difference is geometry: "w/o spherical flow" ignores the spherical constraint and therefore must learn both magnitude and direction, even though the solver depends only on direction. For comparable smoothing, we replace the vMF kernel on $\mathbb{S}^{m-1}$ with a Gaussian kernel in $\mathbb{R}^m$. Fig. 4 (left) shows consistent drops on Dynamic Scheduling and Dynamic Routing, highlighting the value of respecting the solver-induced spherical geometry.

**Smoothed Bellman update.** We next ablate our smoothed Bellman operator, which stabilizes value learning under solver-induced discontinuities. We sweep the smoothing strength $\kappa$ (with `None` denoting no smoothing) and report performance in Fig. 4 (right). Overall, moderate smoothing yields the best returns, while too little or too much smoothing can hurt: weak (or no) smoothing leads to noisy, unstable critic targets, whereas overly strong smoothing oversmooths the value landscape and introduces bias.

**Latent-space critic.** Finally, we evaluate the training-time savings from learning the critic in the latent cost space. As a baseline, we instead train a critic in the action space and use it to update the policy via Eq. (4). Since the action-space critic makes training prohibitively slow, we do not run it to completion; instead, we compare wall-clock time per policy update. Evaluated on the Dynamic Scheduling task, one policy training step takes around 6 minutes with the action-space critic, versus under 1 second with the cost-space critic. This highlights the necessity of training the critic in the cost

space, greatly reducing the number of solver calls.

## 6. Conclusion and Limitations

We introduced LSFLOW, a latent spherical flow policy for RL with combinatorial actions. Our key idea is to shift feasibility enforcement from the policy network to a combinatorial optimization solver, while learning an expressive stochastic policy in a continuous latent space of cost directions. To make learning practical, we amortize solver usage by training both the critic and actor directly in this latent space via a weighted flow-matching objective, and we further stabilize critic learning under solver-induced discontinuities with a smoothed Bellman update. Across combinatorial RL benchmarks and a real-world sexually transmitted infection testing task, LSFLOW achieves strong performance and improved efficiency over prior combinatorial RL methods, suggesting that flow-based generative policies offer a principled and scalable route to RL with combinatorial actions.

**Limitations** We identify several promising directions for future work. First, smoothing introduces a bias–variance trade-off; developing adaptive or state-dependent smoothing schedules could further improve stability without over-smoothing. Second, invoking the solver at inference time incurs additional computational overhead, which remains a common and open challenge for state-of-the-art combinatorial RL methods. Combining our latent-space training with learned guidance (Khalil et al., 2016; Li et al., 2023) could mitigate this cost, further reducing runtime and enabling deployment on larger-scale, real-world constrained decision problems. Third, we optimize the policy by value-weighted flow matching rather than by directly maximizing the critic through the flow sampler; the latter is an appealing alternative but, in our preliminary attempts, backpropagating through the sampler destabilized training in the combinatorial setting. Designing a stable direct value-maximization scheme for this setting is left to future work.

## Acknowledgments

We thank the anonymous reviewers for their valuable feedback. L.K. would like to thank Aaron Ferber for helpful discussions. This work was supported by ONR MURI N00014-24-1-2742.

## Impact Statement

This work advances reinforcement learning for combinatorial action spaces by combining expressive stochastic generative policies with feasibility guarantees from combinatorial optimization, which could improve decision-making in constraint-heavy domains. Potential risks arise when ob-

jectives or constraints are misspecified, potentially leading to unintended or inequitable outcomes. Any real-world use should therefore include careful objective and constraint design (including equity and safety considerations), as well as rigorous auditing under distribution shift and across subpopulations.

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

# Appendix for Latent Spherical Flow Policy for Combinatorial Reinforcement Learning

## A. Limitations

There are several promising directions for future work. First, while we focus on linear-objective solvers, it would be valuable to extend the framework to richer solver interfaces (e.g., parametric constraints, multi-objective solvers, or robust formulations) while preserving feasibility guarantees. Second, smoothing introduces a bias–variance trade-off; developing adaptive or state-dependent smoothing schedules could further improve stability without oversmoothing. Finally, combining our latent-space training with solver warm-starting or learned heuristics may further reduce runtime and enable deployment on larger-scale, real-world constrained decision problems.

## B. Training Algorithm

We provide the full pseudocode of our method in Algorithm 1.

---

**Algorithm 1** Latent Spherical Flow Policy Training

---

1: **Require:** temperature $\lambda$; vMF concentration $\kappa$; discount $\gamma$; smoothing size $J$; target update rate $\eta$; solver $a^\star(\cdot)$
2: **Initialize:** replay buffer $\mathcal{D} \leftarrow \emptyset$; actor $v_\theta$; critic $\widetilde{Q}_\phi$; target $\bar{\phi}$
3: **for** iteration $k = 1, 2, \ldots$ **do**
4:     // *Environment step*
5:     Set $s \leftarrow s'$
6:     Sample cost $c \sim \pi_\theta(\cdot \mid s)$ by integrating Eq. (3)
7:     Sample $\tilde{c} \sim K_\kappa(\cdot \mid c)$                                               (vMF perturbation)
8:     Execute action $a \leftarrow a^\star(s, \tilde{c})$; observe reward $r$ and next state $s'$
9:     Store transition $(s, c, r, s')$ into $\mathcal{D}$
10:     // *Gradient update*
11:     Sample minibatch $\{(s, c, r, s')\} \subset \mathcal{D}$
12:     **Critic target:**
13:     Sample center $c' \sim \pi_\theta(\cdot \mid s')$
14:     Sample perturbations $\{\tilde{c}'^{(j)}\}_{j=1}^J \sim K_\kappa(\cdot \mid c')$
15:     $y_\kappa \leftarrow r + \gamma \frac{1}{J} \sum_{j=1}^J \widetilde{Q}_{\bar{\phi}}(s', \tilde{c}'^{(j)})$
16:     **Critic update:** Minimize $\|\widetilde{Q}_\phi(s, c) - y_\kappa\|_2^2$
17:     **Actor update:**
18:     Sample states $\{s\}$ from $\mathcal{D}$ and draw $c_1 \sim \pi_\theta(\cdot \mid s)$
19:     Compute weight $w(s, c_1) \leftarrow \exp\left(\frac{1}{\lambda} \widetilde{Q}_{\bar{\phi}}(s, c_1)\right)$
20:     Update $\theta$ by minimizing weighted spherical flow matching objective in Eq. (4)
21:     **Target soft update:**
22:     $\bar{\phi} \leftarrow (1 - \eta)\bar{\phi} + \eta\phi$
23: **end for**

---

## C. Additional Related Work

**Combinatorial Optimization Solver-Augmented Neural Networks.** Our work relates to solver-augmented learning, which combines neural networks with *combinatorial optimization (CO) solvers* to ensure predictions satisfy discrete feasibility constraints (Schiffer et al., 2026). The main challenge is end-to-end training through the solver's discrete, non-differentiable argmin. Existing approaches address this using surrogate gradients for black-box solvers (Vlastelica Pogančić et al., 2020), perturbation-based smoothing with implicit differentiation (Berthet et al., 2020), and MIP-specific methods that leverage solver sensitivity information (Ferber et al., 2020). Relatedly, decision-focused learning optimizes prediction models for downstream CO decision quality by placing the CO problem inside the training objective (Wilder et al., 2019; Elmachtoub & Grigas, 2022). These methods primarily target supervised or one-shot decisions, rather than sequential policy learning. Ferber et al. (2024) further embeds CO solvers into generative models (GANs (Goodfellow et al., 2014), VAEs (Kingma & Welling, 2013)) for static structured generation. In contrast, we study *sequential* decision making with an *iteratively* trained generative policy via flow matching, where repeated sampling and solver calls arise during policy evaluation and improvement.

## D. Background on Spherical Flow Matching

We provide details on spherical flow matching used to parameterize the cost-space policy $\pi_\theta(c \mid s)$.

**Background.** Flow matching learns a continuous-time generative model by regressing a parametric vector field toward a known target velocity along paths connecting a simple base distribution to the data distribution (Lipman et al., 2023). In our setting, cost vectors lie on the unit sphere $\mathbb{S}^{m-1}$, which requires respecting the underlying Riemannian geometry.

**Geodesic interpolation on the sphere.** Let $c_0, c_1 \in \mathbb{S}^{m-1}$ denote a base sample and a target sample, respectively. We define the interpolation $\{c_t\}_{t \in [0,1]}$ as the shortest geodesic on the sphere connecting $c_0$ and $c_1$. Let $\theta = \arccos(c_0^\top c_1)$

denote the geodesic distance. The interpolation is given by

$$c_t = \frac{\sin((1-t)\theta)}{\sin\theta}\, c_0 + \frac{\sin(t\theta)}{\sin\theta}\, c_1, \qquad t \in [0,1]. \tag{9}$$

This construction ensures $\|c_t\|_2 = 1$ for all $t$.

**Target tangent velocity.** Differentiating Eq. (9) with respect to $t$ yields a velocity vector that lies in the tangent space of the sphere at $c_t$. We denote this target tangent velocity by $u(c_t, s, t)$. In practice, $u(c_t, s, t)$ depends only on $(c_0, c_1, t)$ and is independent of the policy parameters; see Chen & Lipman (2024) for a closed-form expression.

**Projected vector field parameterization.** We parameterize the policy using a time-dependent vector field $v_\theta(c, s, t) \in \mathbb{R}^m$. To ensure that the induced flow remains on the sphere, we project the vector field onto the tangent space at $c$:

$$\Pi_c = \mathbf{I} - cc^\top. \tag{10}$$

The resulting ODE,

$$\frac{dc_t}{dt} = \Pi_{c_t} v_\theta(c_t, s, t),$$

preserves the unit-norm constraint by construction.

**Sampling.** After training, samples $c_1 \sim \pi_\theta(\cdot \mid s)$ are obtained by integrating the ODE from $t = 0$ to $t = 1$ with initial condition $c_0 \sim p_0$, where $p_0$ is chosen to be the uniform distribution on $\mathbb{S}^{m-1}$ in our implementation.

## E. Proof of Lemma 3.1

**Lemma 3.1** (Positive scale invariance of the solver mapping). *Fix any state $s$ with $\mathcal{A}(s) \neq \emptyset$, and let $a^\star(s, c) = \arg\min_{a \in \mathcal{A}(s)} c^\top a$. Then the solver mapping is positively scale invariant: for any $\alpha > 0$,*

$$a^\star(s, \alpha c) = a^\star(s, c).$$

*Proof.* For any $a \in \mathcal{A}$, $(\alpha c)^\top a = \alpha(c^\top a)$. Since $\alpha > 0$ preserves ordering, $c^\top a_1 \leq c^\top a_2$ if and only if $(\alpha c)^\top a_1 \leq (\alpha c)^\top a_2$. Therefore, the minimizer sets coincide. $\square$

## F. Proof of Proposition 3.2

**Proposition 3.2** (Exact expressivity of solver-induced policies). *Fix a state $s$ and assume the feasible action set is finite. Then for any target distribution $\mu$ over $\mathcal{A}(s)$, there exists a distribution $\pi$ over $\mathbb{S}^{m-1}$ such that if we sample $c \sim \pi$ and take the solver action $a = a^\star(s, c)$, then $a$ is distributed according to $\mu$. Equivalently, for all $a \in \mathcal{A}(s)$,*

$$\mathbb{P}[a^\star(s, c) = a] = \mu(a).$$

*Here, $\mathbb{P}[\cdot]$ denotes probability with respect to the randomness of drawing $c \sim \pi$.*

*Proof.* Fix a state $s$ and write $\mathcal{A} := \mathcal{A}(s)$. Without loss of generality, we represent each feasible action as a binary decision vector $a \in \{0, 1\}^m$, since any bounded integer decision can be encoded in binary (Dantzig, 1963).

For each $a \in \mathcal{A}$ define the action region

$$\mathcal{C}_a := \mathcal{C}_a(s) = \{c \in \mathbb{S}^{m-1} : a^\star(s, c) = a\}.$$

By the (deterministic) tie-breaking in $a^\star$, the sets $\{\mathcal{C}_a\}_{a \in \mathcal{A}}$ form a partition of $\mathbb{S}^{m-1}$. Moreover, since $\mathcal{A}$ is finite and the objective is linear in $c$, each $\mathcal{C}_a$ is Borel measurable.

**Step 1:** $\sigma(\mathcal{C}_{\boldsymbol{a}}) > 0$ **for all** $\boldsymbol{a} \in \mathcal{A}$. Fix any $\boldsymbol{a} \in \mathcal{A}$ and let $P := \mathrm{conv}(\mathcal{A})$. We claim that $\boldsymbol{a}$ is a vertex of $P$. Indeed, suppose $\boldsymbol{a} = \sum_{i=1}^{k} \lambda_i \boldsymbol{a}_i$ for some $\boldsymbol{a}_i \in \mathcal{A}$, $\lambda_i > 0$, $\sum_i \lambda_i = 1$. For any vector $x \in \mathbb{R}^m$, let $x_j$ denote its $j$-th coordinate; in particular, $\boldsymbol{a}_j$ and $(\boldsymbol{a}_i)_j$ denote the $j$-th coordinates of $\boldsymbol{a}$ and $\boldsymbol{a}_i$, respectively. Since every coordinate is binary, for each coordinate $j$: if $\boldsymbol{a}_j = 1$ then $1 = \sum_i \lambda_i (\boldsymbol{a}_i)_j$ forces $(\boldsymbol{a}_i)_j = 1$ for all $i$, and if $\boldsymbol{a}_j = 0$ then $0 = \sum_i \lambda_i (\boldsymbol{a}_i)_j$ forces $(\boldsymbol{a}_i)_j = 0$ for all $i$. Hence all $\boldsymbol{a}_i = \boldsymbol{a}$, so $\boldsymbol{a}$ is an extreme point (vertex) of $P$.

Since $P$ is a polytope and $\boldsymbol{a}$ is a vertex, $\boldsymbol{a}$ is an exposed point; by the separating hyperplane theorem (Boyd & Vandenberghe, 2004) there exists a cost vector $\boldsymbol{c}_0 \in \mathbb{R}^m$ such that $\boldsymbol{a}$ is the *unique* minimizer of $\boldsymbol{c}_0^\top x$ over $x \in P$ (equivalently, of $\boldsymbol{c}_0^\top \boldsymbol{a}'$ over $\boldsymbol{a}' \in \mathcal{A}$). Define the strict-optimality cone

$$K_{\boldsymbol{a}} := \left\{ \boldsymbol{c} \in \mathbb{R}^m : \boldsymbol{c}^\top \boldsymbol{a} < \boldsymbol{c}^\top \boldsymbol{a}' \text{ for all } \boldsymbol{a}' \in \mathcal{A} \setminus \{\boldsymbol{a}\} \right\}.$$

Then $\boldsymbol{c}_0 \in K_{\boldsymbol{a}}$, and $K_{\boldsymbol{a}}$ is an open cone (a finite intersection of open halfspaces). In particular, $K_{\boldsymbol{a}} \cap \mathbb{S}^{m-1}$ is a nonempty open subset of $\mathbb{S}^{m-1}$ and it is contained in $\mathcal{C}_{\boldsymbol{a}}$. Therefore,

$$\sigma(\mathcal{C}_{\boldsymbol{a}}) \geq \sigma\big(K_{\boldsymbol{a}} \cap \mathbb{S}^{m-1}\big) > 0.$$

**Step 2: Mixture over regions matches any target** $\mu$. For each $\boldsymbol{a} \in \mathcal{A}$, define a probability measure $\pi_{\boldsymbol{a}}$ on $\mathbb{S}^{m-1}$ by normalizing surface measure on $\mathcal{C}_{\boldsymbol{a}}$:

$$\pi_{\boldsymbol{a}}(B) := \frac{\sigma\big(B \cap \mathcal{C}_{\boldsymbol{a}}\big)}{\sigma(\mathcal{C}_{\boldsymbol{a}})}, \qquad \text{for any measurable } B \subseteq \mathbb{S}^{m-1}.$$

This is well-defined by Step 1.

Now define the mixture distribution

$$\pi(\cdot) := \sum_{\boldsymbol{a} \in \mathcal{A}} \mu(\boldsymbol{a}) \, \pi_{\boldsymbol{a}}(\cdot).$$

Let $\boldsymbol{c} \sim \pi$. Then, for any $\boldsymbol{a} \in \mathcal{A}$,

$$\mathbb{P}[\boldsymbol{a}^\star(\boldsymbol{s}, \boldsymbol{c}) = \boldsymbol{a}] = \mathbb{P}[\boldsymbol{c} \in \mathcal{C}_{\boldsymbol{a}}] = \pi(\mathcal{C}_{\boldsymbol{a}}) = \sum_{\boldsymbol{a}' \in \mathcal{A}} \mu(\boldsymbol{a}') \, \pi_{\boldsymbol{a}'}(\mathcal{C}_{\boldsymbol{a}}) = \mu(\boldsymbol{a}),$$

since $\pi_{\boldsymbol{a}'}(\mathcal{C}_{\boldsymbol{a}}) = \mathbb{1}\{\boldsymbol{a}' = \boldsymbol{a}\}$. $\qquad \square$

## G. Proof of Theorem 3.5

**Theorem 3.5** (Contraction and smoothness of the fixed point). *Let $\mathcal{B}_\infty$ be the space of bounded functions $Q : \mathcal{S} \times \mathbb{S}^{m-1} \to \mathbb{R}$ equipped with the sup norm $\|Q\|_\infty := \sup_{\boldsymbol{s} \in \mathcal{S}, \, \boldsymbol{c} \in \mathbb{S}^{m-1}} |Q(\boldsymbol{s}, \boldsymbol{c})|$. For a fixed $\boldsymbol{s}$, the notation $Q(\boldsymbol{s}, \cdot) \in C^\infty(\mathbb{S}^{m-1})$ means that the map $\boldsymbol{c} \mapsto Q(\boldsymbol{s}, \boldsymbol{c})$ is infinitely differentiable on $\mathbb{S}^{m-1}$. Under Assumption 3.4 with $0 \leq \gamma < 1$, and using the vMF kernel $K_\kappa$ in Eq. (6), the following hold:*

1. *($\gamma$-contraction) $\mathcal{T}_\kappa^\pi$ is a $\gamma$-contraction on $(\mathcal{B}_\infty, \| \cdot \|_\infty)$ and admits a unique fixed point $Q_\kappa^\pi$.*

2. *($C^\infty$ smoothing in $\boldsymbol{c}$) For any $Q \in \mathcal{B}_\infty$ and any $\boldsymbol{s}$, $(\mathcal{T}_\kappa^\pi Q)(\boldsymbol{s}, \cdot) \in C^\infty(\mathbb{S}^{m-1})$.*

3. *(Regularity of the fixed point) Consequently, for every $\boldsymbol{s}$, $Q_\kappa^\pi(\boldsymbol{s}, \cdot) \in C^\infty(\mathbb{S}^{m-1})$.*

*Proof of Theorem 3.5.* Recall the smoothed Bellman operator in Eq. (7):

$$(\mathcal{T}_\kappa^\pi Q)(\boldsymbol{s}, \boldsymbol{c}) = \mathbb{E}\big[ r\big(\boldsymbol{s}, \boldsymbol{a}^\star(\boldsymbol{s}, \tilde{\boldsymbol{c}})\big) + \gamma \, Q(\boldsymbol{s}', \tilde{\boldsymbol{c}}') \big],$$

where $\tilde{\boldsymbol{c}} \sim K_\kappa(\cdot \mid \boldsymbol{c})$, $\boldsymbol{s}' \sim P(\cdot \mid \boldsymbol{s}, \boldsymbol{a}^\star(\boldsymbol{s}, \tilde{\boldsymbol{c}}))$, $\boldsymbol{c}' \sim \pi(\cdot \mid \boldsymbol{s}')$, and $\tilde{\boldsymbol{c}}' \sim K_\kappa(\cdot \mid \boldsymbol{c}')$.

**(1) $\gamma$-contraction and uniqueness.** *Self-map.* For any $Q \in \mathcal{B}_\infty$ and any $(s, c)$,

$$|(\mathcal{T}_\kappa^\pi Q)(s, c)| \leq \mathbb{E}[|r(s, a^\star(s, \tilde{c}))|] + \gamma \, \mathbb{E}[|Q(s', \tilde{c}')|] \leq \|r\|_\infty + \gamma \|Q\|_\infty,$$

so $\mathcal{T}_\kappa^\pi Q$ is bounded and hence $\mathcal{T}_\kappa^\pi : \mathcal{B}_\infty \to \mathcal{B}_\infty$.

*Completeness.* Equipped with the sup norm, $(\mathcal{B}_\infty, \|\cdot\|_\infty)$ is a Banach space.

Let $Q_1, Q_2 \in \mathcal{B}_\infty$. The reward terms cancel, so

$$(\mathcal{T}_\kappa^\pi Q_1)(s, c) - (\mathcal{T}_\kappa^\pi Q_2)(s, c) = \gamma \, \mathbb{E}[Q_1(s', \tilde{c}') - Q_2(s', \tilde{c}')].$$

Taking absolute values and using the definition of $\|\cdot\|_\infty$,

$$\left|(\mathcal{T}_\kappa^\pi Q_1)(s, c) - (\mathcal{T}_\kappa^\pi Q_2)(s, c)\right| \leq \gamma \|Q_1 - Q_2\|_\infty.$$

Taking the supremum over $(s, c)$ yields

$$\|\mathcal{T}_\kappa^\pi Q_1 - \mathcal{T}_\kappa^\pi Q_2\|_\infty \leq \gamma \|Q_1 - Q_2\|_\infty,$$

so $\mathcal{T}_\kappa^\pi$ is a $\gamma$-contraction on $(\mathcal{B}_\infty, \|\cdot\|_\infty)$. Since $0 \leq \gamma < 1$, the Banach fixed-point theorem implies that $\mathcal{T}_\kappa^\pi$ admits a unique fixed point $Q_\kappa^\pi \in \mathcal{B}_\infty$.

**(2) $C^\infty$ smoothing in $c$.** Fix any $s$ and any bounded $Q \in \mathcal{B}_\infty$. Define the bounded measurable function

$$G_s(\tilde{c}) := \mathbb{E}\left[r\big(s, a^\star(s, \tilde{c})\big) + \gamma \, Q(s', \tilde{c}') \,\big|\, \tilde{c}\right],$$

where the conditional expectation is over $s' \sim P(\cdot \mid s, a^\star(s, \tilde{c}))$, $c' \sim \pi(\cdot \mid s')$, and $\tilde{c}' \sim K_\kappa(\cdot \mid c')$. By Assumption 3.4, $r$ and $Q$ are bounded, hence

$$|G_s(\tilde{c})| \leq \|r\|_\infty + \gamma \|Q\|_\infty \quad \text{for all } \tilde{c} \in \mathbb{S}^{m-1}.$$

Using $\tilde{c} \sim K_\kappa(\cdot \mid c)$, we rewrite the operator as

$$(\mathcal{T}_\kappa^\pi Q)(s, c) = \int_{\mathbb{S}^{m-1}} G_s(\tilde{c}) \, K_\kappa(\tilde{c} \mid c) \, d\sigma(\tilde{c}), \tag{11}$$

where $d\sigma$ is the surface measure on $\mathbb{S}^{m-1}$. By the vMF form in Eq. (6), the kernel $K_\kappa(\tilde{c} \mid c)$ is $C^\infty$ in $c$ for every fixed $\tilde{c}$. Moreover, since $\mathbb{S}^{m-1} \times \mathbb{S}^{m-1}$ is compact, all derivatives of $K_\kappa(\tilde{c} \mid c)$ with respect to $c$ are bounded uniformly over $(\tilde{c}, c)$.

Therefore, fix any smooth chart $\varphi : U \subset \mathbb{S}^{m-1} \to V \subset \mathbb{R}^{m-1}$ and write $c = \varphi^{-1}(z)$ for $z \in V$. For any multi-index $\alpha$, consider the partial derivative $\partial_z^\alpha$ of the coordinate representation

$$F(z) := (\mathcal{T}_\kappa^\pi Q)\big(s, \varphi^{-1}(z)\big) = \int_{\mathbb{S}^{m-1}} G_s(\tilde{c}) \, K_\kappa\big(\tilde{c} \mid \varphi^{-1}(z)\big) \, d\sigma(\tilde{c}).$$

Since $K_\kappa(\tilde{c} \mid c)$ is $C^\infty$ in $c$ and $\varphi^{-1}$ is smooth, the map $z \mapsto K_\kappa\big(\tilde{c} \mid \varphi^{-1}(z)\big)$ is $C^\infty$ on $V$ for each fixed $\tilde{c}$. Moreover, because $\mathbb{S}^{m-1} \times \mathbb{S}^{m-1}$ is compact, all derivatives $\partial_z^\alpha K_\kappa\big(\tilde{c} \mid \varphi^{-1}(z)\big)$ are bounded uniformly over $\tilde{c} \in \mathbb{S}^{m-1}$ and $z$ in compact subsets of $V$. Hence, using the bound on $G_s$ and dominated convergence, we may differentiate under the integral sign to obtain

$$\partial_z^\alpha F(z) = \int_{\mathbb{S}^{m-1}} G_s(\tilde{c}) \, \partial_z^\alpha K_\kappa\big(\tilde{c} \mid \varphi^{-1}(z)\big) \, d\sigma(\tilde{c}),$$

and the right-hand side is continuous in $z$. Since this holds for all charts $\varphi$ and all multi-indices $\alpha$, we conclude that $(\mathcal{T}_\kappa^\pi Q)(s, \cdot) \in C^\infty(\mathbb{S}^{m-1})$.

**(3) Regularity of the fixed point.** From Part (1), the unique fixed point satisfies $Q_\kappa^\pi = \mathcal{T}_\kappa^\pi Q_\kappa^\pi$ and $Q_\kappa^\pi \in \mathcal{B}_\infty$. Applying Part (2) with $Q = Q_\kappa^\pi$ yields $Q_\kappa^\pi(s, \cdot) \in C^\infty(\mathbb{S}^{m-1})$ for every $s$. $\qquad\square$

**Approximate solvers.** Exact inner optimality is not required for Theorem 3.5. Suppose the exact solver $a^\star$ is replaced by any feasible, measurable map $\hat{a}(s, c) \in \mathcal{A}(s)$, and the smoothed operator $\mathcal{T}_\kappa^\pi$ in (7) is defined with $\hat{a}$ in place of $a^\star$. The contraction argument in Part (1) uses only boundedness of the reward and $0 \leq \gamma < 1$, and the smoothing argument in Part (2) uses only that the kernel $K_\kappa(\tilde{c} \mid c)$ is $C^\infty$ in $c$; neither uses optimality of $a^\star$. Hence the same conclusions hold: the modified operator is a $\gamma$-contraction with a unique fixed point, and that fixed point is $C^\infty$ in $c$. Only the interpretation changes—the fixed point is the smoothed value induced by the approximate solver $\hat{a}$ rather than by the exact argmin.

## H. Proof of Theorem 3.6

**Theorem 3.6** (Consistency of the smoothed fixed point). *For each reachable state $s$, let $\mathcal{B}_s := \{c \in \mathbb{S}^{m-1} :$ $\arg\min_{a \in \mathcal{A}(s)} c^\top a$ is not a singleton$\}$ denote the solver-switching boundary, and assume $\pi(\cdot \mid s)$ assigns zero mass to $\mathcal{B}_s$. Then for every reachable $s$ and every $c \notin \mathcal{B}_s$,*

$$Q_\kappa^\pi(s, c) \longrightarrow Q^\pi(s, c) \qquad as \ \kappa \to \infty,$$

*where $Q^\pi$ is the fixed point of the unsmoothed Bellman operator $\mathcal{T}^\pi$. Since $\mathcal{B}_s$ has surface measure zero, this convergence also holds for surface-a.e. $c \in \mathbb{S}^{m-1}$, and $Q_\kappa^\pi(s, C) \to Q^\pi(s, C)$ almost surely when $C \sim \pi(\cdot \mid s)$.*

*Proof.* Throughout, write $R_{\max} := \|r\|_\infty < \infty$ (Assumption 3.4) and recall that $\mathcal{A}(s)$ is finite for every reachable $s$. Both fixed points are bounded by $R_{\max}/(1 - \gamma)$: from $Q_\kappa^\pi = \mathcal{T}_\kappa^\pi Q_\kappa^\pi$ we get $\|Q_\kappa^\pi\|_\infty \leq R_{\max} + \gamma \|Q_\kappa^\pi\|_\infty$, and likewise for $Q^\pi$.

We first record a local-constancy property of the solver away from the switching boundary and a two-step concentration property of the smoothing kernel, then couple the smoothed and unsmoothed evaluations and control their gap through a first-disagreement time.

**Preliminary 1: local constancy of the solver.** Fix a reachable $s$ and a direction $c \notin \mathcal{B}_s$. The minimizer $a^\star(s, c)$ is then unique, and the margin

$$\delta_s(c) := \min_{a \in \mathcal{A}(s) \setminus \{a^\star(s,c)\}} c^\top\big(a - a^\star(s, c)\big) > 0.$$

Let $D(s) := \max_{a, a' \in \mathcal{A}(s)} \|a - a'\|_2$. For any $\hat{c} \in \mathbb{S}^{m-1}$ and any suboptimal $a$,

$$\hat{c}^\top\big(a - a^\star(s, c)\big) = c^\top\big(a - a^\star(s, c)\big) + (\hat{c} - c)^\top\big(a - a^\star(s, c)\big) \geq \delta_s(c) - D(s) \|\hat{c} - c\|_2.$$

Hence $\|\hat{c} - c\|_2 < \delta_s(c)/D(s)$ implies

$$a^\star(s, \hat{c}) = a^\star(s, c), \tag{12}$$

so $a^\star(s, \cdot)$ is locally constant at every $c \notin \mathcal{B}_s$. The kernels below are absolutely continuous with respect to surface measure, and the tie surfaces have measure zero, so the perturbed directions avoid $\mathcal{B}_s$ almost surely and all solver outputs below are a.s. well defined.

**Preliminary 2: one- and two-step concentration.** For a center $x \in \mathbb{S}^{m-1}$ and $Z \sim K_\kappa(\cdot \mid x)$, the rotational symmetry of the vMF kernel (6) makes the law of $x^\top Z$ independent of the center $x$, so the one-step tail

$$p_\kappa(r) := \sup_{x \in \mathbb{S}^{m-1}} \mathbb{P}_{Z \sim K_\kappa(\cdot \mid x)}\big(\|Z - x\|_2 \geq r\big)$$

does not depend on the center. Since $\|Z - x\|_2^2 = 2(1 - x^\top Z)$ with $1 - x^\top Z \geq 0$, Markov's inequality gives, for every fixed $r > 0$,

$$p_\kappa(r) = \mathbb{P}\big(1 - x^\top Z \geq r^2/2\big) \leq \frac{2\,\mathbb{E}[1 - x^\top Z]}{r^2}.$$

The vMF kernel concentrates at its center as $\kappa \to \infty$, so $\mathbb{E}[x^\top Z] \to 1$ and hence $\mathbb{E}[1 - x^\top Z] \to 0$, giving

$$p_\kappa(r) \xrightarrow[\kappa \to \infty]{} 0. \tag{13}$$

At non-initial steps the bootstrap direction is perturbed once and then again before the next solve, so the relevant object is the two-step kernel

$$L_\kappa(\cdot \mid c) := \int_{\mathbb{S}^{m-1}} K_\kappa(\cdot \mid x)\, K_\kappa(dx \mid c),$$

the law of $Y$ obtained by $X \sim K_\kappa(\cdot \mid c), Y \sim K_\kappa(\cdot \mid X)$. By the triangle inequality $\|Y - c\|_2 \leq \|Y - X\|_2 + \|X - c\|_2$ and a union bound, for every $r > 0$,

$$q_\kappa(r) := \sup_{x \in \mathbb{S}^{m-1}} \mathbb{P}_{Y \sim L_\kappa(\cdot \mid x)}\big(\|Y - x\|_2 \geq r\big) \leq 2\, p_\kappa(r/2) \xrightarrow[\kappa \to \infty]{} 0, \tag{14}$$

using that, conditional on $X = x'$, the law of $Y$ is $K_\kappa(\cdot \mid x')$, hence $\mathbb{P}(\|Y - X\|_2 \geq r/2 \mid X = x') \leq p_\kappa(r/2)$ for every $x'$.

**Step 1: coupled evaluations and the first-disagreement time.** Fix $s$ and $c \notin \mathcal{B}_s$. Unrolling $Q^\pi = \mathcal{T}^\pi Q^\pi$ and $Q_\kappa^\pi = \mathcal{T}_\kappa^\pi Q_\kappa^\pi$ for $T$ steps generates two processes on $\mathcal{S} \times \mathbb{S}^{m-1}$, both started at $(s_0, c_0) = (s, c)$, with on-policy centers $c_t \sim \pi(\cdot \mid s_t)$ for $t \geq 1$. The unsmoothed process solves at the center, $a_t^0 = a^\star(s_t^0, c_t^0)$; the smoothed process solves at a perturbed direction $\tilde{c}_t$, where $\tilde{c}_0 \sim K_\kappa(\cdot \mid c_0)$ and, for $t \geq 1$, $\tilde{c}_t \sim L_\kappa(\cdot \mid c_t^\kappa)$, giving $a_t^\kappa = a^\star(s_t^\kappa, \tilde{c}_t)$. Couple the processes so that equal actions share transition noise and equal next states share the next center draw. As long as the actions have agreed the states and centers agree too; in particular, on the event $\{\tau_\kappa \geq t\}$ we have $(s_t^\kappa, c_t^\kappa) = (s_t^0, c_t^0)$. Define the first-disagreement time

$$\tau_\kappa := \inf\{t \geq 0 : a_t^\kappa \neq a_t^0\}.$$

With $G_T^0 := \sum_{t=0}^{T-1} \gamma^t r(s_t^0, a_t^0)$ and $G_T^\kappa := \sum_{t=0}^{T-1} \gamma^t r(s_t^\kappa, a_t^\kappa)$, the event $\{\tau_\kappa \geq T\}$ forces $G_T^\kappa = G_T^0$, so

$$\big|\mathbb{E}[G_T^\kappa] - \mathbb{E}[G_T^0]\big| \leq \frac{2R_{\max}}{1 - \gamma} \mathbb{P}(\tau_\kappa < T). \tag{15}$$

**Step 2: the disagreement probability vanishes at every fixed horizon.** At $t = 0$ the processes share $(s, c)$ with $c \notin \mathcal{B}_s$, so by (12) a disagreement needs $\|\tilde{c}_0 - c\|_2 \geq \delta_s(c)/D(s)$, whence $\mathbb{P}(\tau_\kappa = 0) \leq p_\kappa(\delta_s(c)/D(s)) \to 0$ by (13). For $t \geq 1$, set

$$f_\kappa(s, c) := \mathbb{P}_{Y \sim L_\kappa(\cdot \mid c)}\big(a^\star(s, Y) \neq a^\star(s, c)\big) \in [0, 1].$$

By (12) and (14), $f_\kappa(s, c) \leq q_\kappa(\delta_s(c)/D(s)) \to 0$ for every $c \notin \mathcal{B}_s$. On $\{\tau_\kappa \geq t\}$ the coupling gives $(s_t^\kappa, c_t^\kappa) = (s_t^0, c_t^0)$ and $\tilde{c}_t \sim L_\kappa(\cdot \mid c_t^0)$, so

$$\mathbb{P}(\tau_\kappa = t) \leq \mathbb{E}\big[f_\kappa(s_t^0, c_t^0)\big].$$

This expectation is under the *unsmoothed* marginal of $(s_t^0, c_t^0)$, which is independent of $\kappa$. The zero-boundary-mass assumption gives $c_t^0 \notin \mathcal{B}_{s_t^0}$ a.s., so $f_\kappa(s_t^0, c_t^0) \to 0$ a.s.; since $f_\kappa \leq 1$, dominated convergence yields $\mathbb{P}(\tau_\kappa = t) \to 0$ for each fixed $t \geq 1$. Summing the finitely many disjoint events,

$$\mathbb{P}(\tau_\kappa < T) = \sum_{t=0}^{T-1} \mathbb{P}(\tau_\kappa = t) \xrightarrow[\kappa \to \infty]{} 0 \qquad \text{for every fixed } T.$$

**Step 3: removing the truncation.** Using $\|Q^\pi\|_\infty, \|Q_\kappa^\pi\|_\infty \leq R_{\max}/(1 - \gamma)$, the discounted tails obey $\big|Q^\pi(s, c) - \mathbb{E}[G_T^0]\big| \leq R_{\max}\gamma^T/(1 - \gamma)$ and similarly for $Q_\kappa^\pi$. Combining with (15),

$$\big|Q_\kappa^\pi(s, c) - Q^\pi(s, c)\big| \leq \frac{2R_{\max}\gamma^T}{1 - \gamma} + \frac{2R_{\max}}{1 - \gamma} \mathbb{P}(\tau_\kappa < T).$$

Given $\varepsilon > 0$, pick $T$ making the first term $\leq \varepsilon/2$, then $\kappa$ large enough (Step 2) making the second $\leq \varepsilon/2$. Thus $Q_\kappa^\pi(s, c) \to Q^\pi(s, c)$ for every reachable $s$ and every $c \notin \mathcal{B}_s$.

**Step 4: a.e. and a.s. statements.** Each tie surface $\{c : c^\top(a - a') = 0\}$ has surface measure zero, and $\mathcal{B}_s$ is contained in their finite union over $a \neq a' \in \mathcal{A}(s)$; hence $\mathcal{B}_s$ has surface measure zero and the convergence holds for surface-a.e. $c \in \mathbb{S}^{m-1}$. Finally, since $\pi(\cdot \mid s)$ assigns zero mass to $\mathcal{B}_s$, a draw $C \sim \pi(\cdot \mid s)$ lies outside $\mathcal{B}_s$ a.s., giving $Q_\kappa^\pi(s, C) \to Q^\pi(s, C)$ almost surely. $\qquad \square$

# I. Weighted Flow Matching = KL-Regularized Policy Improvement

**Theorem I.1** (Weighted flow matching implements KL-regularized policy improvement). *Assume $w_k(s, c) \geq 0$ and for $D$-a.e. $s$,*

$$0 < Z_k(s) := \mathbb{E}_{c \sim \pi_k(\cdot|s)}[w_k(s, c)] < \infty. \tag{16}$$

*Define the reweighted distribution*

$$\pi_{k+1}(c \mid s) := \frac{1}{Z_k(s)} \pi_k(c \mid s)\, w_k(s, c). \tag{17}$$

*Then:*

*(i) Minimizing the weighted flow-matching objective Eq. (4) is equivalent (i.e., has the same minimizers) to minimizing the unweighted flow-matching objective with $c_1 \sim \pi_{k+1}(\cdot \mid s)$:*

$$\tilde{\mathcal{L}}_k(\theta) := \mathbb{E}\Big[\big\|\Pi_{c_t} v_\theta(c_t, s, t) - u(c_t, s, t)\big\|_2^2\Big], \tag{18}$$

*where the expectation is over $s \sim D$, $c_1 \sim \pi_{k+1}(\cdot \mid s)$, $c_0 \sim p_0$, and $t \sim U(0, 1)$.*

*(ii) With the exponential weighting $w_k(s, c) \propto \exp\big(\frac{1}{\lambda} Q(s, a^*(s, c))\big)$, the distribution $\pi_{k+1}(\cdot \mid s)$ in Eq. (17) coincides with the solution of the KL-regularized policy improvement problem*

$$\pi_{k+1}(\cdot \mid s) \in \arg\max_{\pi(\cdot|s)} \mathbb{E}_{c \sim \pi(\cdot|s)}\big[Q(s, a^*(s, c))\big] - \lambda\, \mathrm{KL}\big(\pi(\cdot \mid s) \,\|\, \pi_k(\cdot \mid s)\big), \tag{19}$$

*whose closed form is the exponential tilt $\pi_{k+1}(c \mid s) \propto \pi_k(c \mid s) \exp\big(\frac{1}{\lambda} Q(s, a^*(s, c))\big)$.*

*Proof.* Fix $s$ and define the conditional regression loss

$$g_\theta(s, c_1) := \mathbb{E}_{c_0 \sim p_0,\, t \sim U(0,1)}\Big[\big\|\Pi_{c_t} v_\theta(c_t, s, t) - u(c_t, s, t)\big\|_2^2\Big],$$

so that Eq. (4) can be written as

$$\mathcal{L}_k(\theta) = \mathbb{E}_{s \sim D}\Big[\mathbb{E}_{c_1 \sim \pi_k(\cdot|s)}\big[w_k(s, c_1)\, g_\theta(s, c_1)\big]\Big].$$

**(i)** By the definition of $\pi_{k+1}$ in Eq. (17), for each fixed $s$ we have

$$\mathbb{E}_{c_1 \sim \pi_k(\cdot|s)}\big[w_k(s, c_1)\, g_\theta(s, c_1)\big] = Z_k(s)\, \mathbb{E}_{c_1 \sim \pi_{k+1}(\cdot|s)}\big[g_\theta(s, c_1)\big].$$

Plugging this into the outer expectation gives

$$\mathcal{L}_k(\theta) = \mathbb{E}_{s \sim D}\Big[Z_k(s)\, \mathbb{E}_{c_1 \sim \pi_{k+1}(\cdot|s)}\big[g_\theta(s, c_1)\big]\Big].$$

Since $Z_k(s) > 0$ and does not depend on $\theta$, multiplying the per-$s$ objective by $Z_k(s)$ does not change the set of minimizers over $\theta$. This shows that minimizing $\mathcal{L}_k(\theta)$ is equivalent to minimizing $\tilde{\mathcal{L}}_k(\theta)$ in Eq. (18).

**(ii)** With $w_k(s, c) \propto \exp\big(\frac{1}{\lambda} Q(s, a^*(s, c))\big)$, the reweighted distribution Eq. (17) becomes

$$\pi_{k+1}(c \mid s) \propto \pi_k(c \mid s) \exp\Big(\frac{1}{\lambda} Q(s, a^*(s, c))\Big),$$

which matches the standard closed-form solution of the KL-regularized update in Eq. (19). □

# J. Experimental Details of Benchmarks

## J.1. Environment Details

**Dynamic scheduling.** Consider the example of scheduling $N$ volunteer health workers to $J$ patients, where each have limited time windows for scheduling and $K$ available timeslots. An action here corresponds to selecting up to $B$ patients to

serve and assigning workers to these patients during feasible time slots. As defined in the benchmark (Xu et al., 2025), the scheduling problem can be formulated as

$$
\begin{aligned}
\max_{x,a} \quad & Q(s,a) && (20)\\
\text{s.t.} \quad & \sum_{j \in [J]} a_j \leq B, \\
& \sum_{j \in [J]} x_{ijk} \leq 1 && \forall i \in [N],\ k \in [K], \\
& \sum_{i \in [N],\, k \in [K]} x_{ijk} \leq 2 && \forall j \in [J], \\
& 2\, a_j \leq \sum_{i \in [N],\, k \in [K]} x_{ijk} && \forall j \in [J], \\
& a_j \geq x_{ijk} && \forall i \in [N],\ j \in [J],\ k \in [K], \\
& a_j \in \{0,1\} && \forall j \in [J], \\
& x_{ijk} \in \{0,1\} && \forall i \in [N],\ j \in [J],\ k \in [K].
\end{aligned}
$$

We have a decision variable $x_{ijk}$ whenever a worker $i$ and patient $j$ are both available at timeslot $k$.

**Dynamic routing.** States lie on nodes of a graph, and an action corresponds to a bounded-length route that starts and ends at a designated spot. For this problem, we use the graph of the real-world network of the London underground[2]. One node is placed at each station.

We model this routing problem as a constrained action-selection on a graph $G(\mathcal{V}, \mathcal{E})$ with a designated source $\hat{v} \in \mathcal{V}$. The agent must select a path that starts and ends at this source while also ensuring that this stays within the budget of the number of nodes we can act on. We set the maximum length of this path, $T$, to be double this budget, or $T = 2B$. We then use a flow variable $f_{j,k,t} \in \{0,1\}$, where $f_{j,k,t} = 1$ indicates that at timestep $t$, the path covers edge $(j,k)$. A separate variable, $a_j$ indicates whether we pass through a node $j$.

$$
\begin{aligned}
\max_{\{f_{j,k,t}\}, \{a_j\}} \quad & Q(s,a) \\
\text{s.t.} \quad & \sum_{k:(\hat{v},k)\in\mathcal{E}} f_{\hat{v},k,1} = 1 \\
& \sum_{k:(k,\hat{v})\in\mathcal{E}} f_{k,\hat{v},T} = 1 \\
& \sum_{(j,k)\in\mathcal{E}} f_{j,k,t} = 1, && \forall t \in \{1,\ldots,T\} \\
& \sum_{k:(k,j)\in\mathcal{E}} f_{k,j,t} = \sum_{k:(j,k)\in\mathcal{E}} f_{j,k,t+1}, && \forall j \in \mathcal{V},\ \forall t \in \{1,\ldots,T-1\} \quad (21)\\
& a_j \leq \sum_{t\in\{1,\ldots,T\}} \sum_{k:(k,j)\in\mathcal{E}} f_{j,k,t}, && \forall j \in \mathcal{V} \\
& \sum_{j\in\mathcal{V}} a_j \leq B \\
& f_{j,k,t} \in \{0,1\}, && \forall j,k \in \mathcal{V},\ \forall t \in \{1,\ldots,T\} \\
& a_j \in \{0,1\}, && \forall j \in \mathcal{V}.
\end{aligned}
$$

---

[2]`https://commons.wikimedia.org/wiki/London_Underground_geographic_maps`

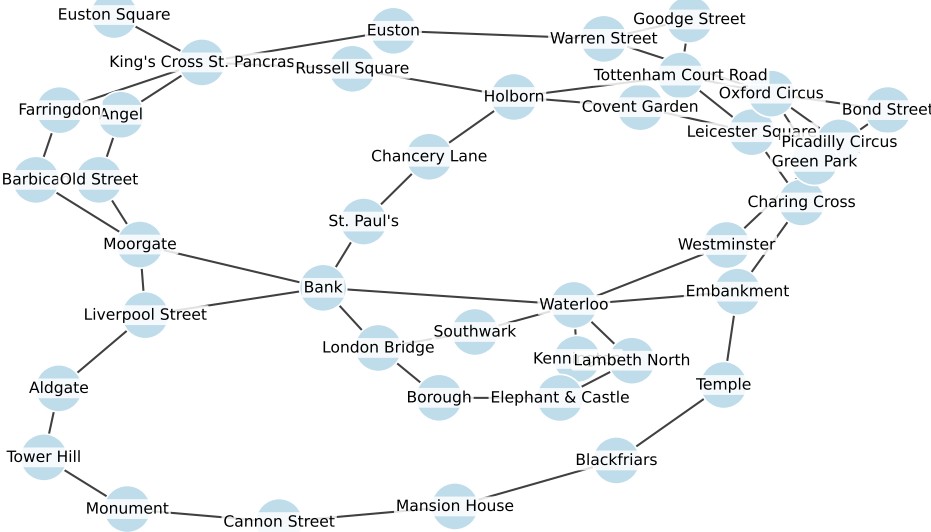

*Figure 5.* Graph used for the dynamic routing problem, based on the London tube network.

**Dynamic assignment.** This problem is an instance of the NP-hard generalized assignment problem (Özbakir et al., 2010). In this setting, workers have a given capacity $b_i$, where serving patient $j$ has a cost $c_j$. An action assigns workers to states without exceeding any worker's capacity. We have a decision variable $x_{ij}$ for all $i$ and $j$ that indicates whether worker $i$ gets assigned to state $j$.

$$
\max_{x,a} \quad Q(s,a) \tag{22}
$$
$$
\text{s.t.} \quad \sum_{j \in [J]} c_j x_{ij} \leq b_i \qquad\qquad \forall i \in [N],
$$
$$
a_j \leq \sum_{i \in [N]} x_{ij} \qquad\qquad \forall j \in [J],
$$
$$
x_{ij} \in \{0,1\} \qquad\qquad \forall i \in [N],\ j \in [J],
$$
$$
a_j \geq x_{ij} \qquad\qquad \forall i \in [N],\ j \in [J],
$$
$$
a_j \in \{0,1\} \qquad\qquad \forall j \in [J].
$$

**Dynamic intervention.** In public health programs, different interventions may impact different groups of individuals. We model this as a finite-horizon reinforcement learning problem. In this problem, each action $a_i \in \{0,1\}$ corresponds to one of $N$ possible interventions. In the case of health workers and patients, consider the patient $j$ and their state $s_j$ measuring their engagement level. Each action has a cost $c_i$, and we are limited by a total budget $B$, producing the optimization problem

$$
\max_a \ Q(s,a) \quad \text{s.t.} \quad \sum_{j \in [J]} c_j a_j \leq B, \qquad a_j \in \{0,1\}\ \forall j \in [J]. \tag{23}
$$

Following the baseline method, we model the probability patient $j$ transitions from their current engagement state $s_j$ to a higher ($s_j^+$) or lower ($s_j^-$) state as follows

$$
P_j(s_j, a, s_j^+) = \phi_j\big(\omega_j(s_j)^\top a\big), \qquad P_j(s_j, a, s_j^-) = 1 - P_j(s_j, a, s_j^+).
$$

where $w_j(s_j) \in \mathbb{R}^N$ represents how interventions affect $j$ potentially dependent on their current state.

In many real-world public health settings, the effect of additional actions is nonlinear. For example, the widely used model of smoking cessation is described as "S-shape", suggesting that earlier interventions can increase an individual's chances of improvement, with later interventions only resulting in a plateau (Levy et al., 2006). Following (Xu et al., 2025), we use these as motivating descriptions for the link function between the action score and the upward transition probability. For each state $j$, we form a score $x$ and map it to a probability with a sigmoid link

$$\tilde{\phi}_j(x) = a_j \, \sigma(b_j x - x_{0,j}), \qquad \sigma(u) = \frac{1}{1 + e^{-u}}, \tag{24}$$

We partition states into two subsets with different sigmoid response parameters. For one, we sample $a \in [0.7, 1.0]$, $b \in [1.4, 1.9]$, $x_0 \in \{0, 10\}$, and for the other we sample $a \in [0.2, 0.5]$, set $b = 1$, and sample $x_0 \in \{1, 2\}$.

In implementation, we use a stabilized version

$$\phi_j(x) = \alpha\big(\tilde{\phi}_j(x) - \tilde{\phi}_j(0)\big) + \varepsilon, \tag{25}$$

with $\alpha = 0.98$ and $\varepsilon = 0.01$.

### J.2. Implementation Details

**Benchmark setup.** Across all benchmark settings, we use population size $N = 40$, budget $B = 10$, and horizon $H = 20$.

**Training details of LSFLOW.** We follow Ma et al. (2025a) and incorporate two stabilizing steps. Before executing an action, rather than sampling a single cost direction, we draw multiple candidate directions and select the one with the highest critic value. We also use an exponential moving average when forming the flow-matching weights during training

Our method is implemented in PyTorch with TorchRL (Bou et al., 2023). Hyperparameters are listed in Table 2.

**Compared methods.** DQN-Sampling (He et al., 2016) samples 100 feasible actions at each decision point and executes the one with the highest critic value. We use the authors' original implementation[3] of SEQUOIA (Xu et al., 2025). SRL was originally implemented in Julia[4]; we re-implemented it in PyTorch for a fair comparison. For a fair comparison, all RL methods use the same critic-network architecture and the same number of warm-up steps, and are then trained until convergence.

**Heuristic baselines.** We additionally evaluate simple *Random* and *Greedy* heuristics tailored to each environment:

- **Dynamic Scheduling.** *Random* randomly permutes workers and patients, then iterates over patients and assigns the first two compatible unassigned workers; it stops when no feasible assignment remains. *Greedy* repeatedly selects the unassigned patient with the largest number of compatible unassigned workers and assigns two compatible workers, until the budget is exhausted or no feasible assignment remains.

- **Dynamic Routing.** *Random* enumerates simple cycles of length $T = 2B$ using `networkx.simple_cycles`, uniformly samples one cycle, and then uniformly samples $B$ nodes on that cycle to act on. Following Xu et al. (2025), we restrict to *simple* cycles. This is weaker than uniformly sampling a feasible action, but implementing the latter would be substantially more complex. *Greedy* selects the longest feasible cycle and then chooses $B$ nodes along it.

- **Dynamic Assignment.** *Random* repeatedly assigns workers to states uniformly at random among the assignments that respect the remaining worker capacities, until no feasible assignment remains. *Greedy* instead assigns workers to states in a greedy order while respecting capacity constraints, until no feasible assignment remains.

- **Dynamic Intervention.** *Random* uniformly samples a subset of size $B$. *Greedy* uses a warmup-then-exploit strategy: for the first 5 steps it executes random feasible actions while tracking empirical mean rewards; afterward, it selects the $B$ actions with the highest observed average reward at each step.

---

[3] `https://github.com/lily-x/combinatorial-rmab`
[4] `https://github.com/tumBAIS/Structured-RL`

*Table 2.* Hyperparameters used for benchmarks.

| Hyperparameter | Setting |
|---|---|
| *Training schedule* | |
| Warmup steps | 1000 |
| Training episodes (`train_updates`) | 2000 |
| Batch size | 64 |
| *Core RL / optimization* | |
| Discount factor ($\gamma$) | 0.99 |
| Learning rate (Adam) | $1 \times 10^{-3}$ |
| Target update rate ($\tau$) | 0.005 |
| Delayed update | 2 |
| Reward scaling | 1.0 |
| Gradient clipping (max norm) | 5.0 |
| *Policy training* | |
| Number of particles ($K$) | 12 |
| $\lambda$ schedule | start 2.0, end 0.8, steps 10000 |
| Weight clip ($w_{\max}$) | 4.0 |
| vMF concentration ($\kappa$) | 28.0 |
| vMF perturbations per candidate ($J$) | 1 |
| Flow steps | 36 |
| Q-norm clip | 3.0 |
| Q running beta | 0.05 |
| *Policy network (MLP)* | |
| Time embedding | 32-d Fourier features ($L = 16$ harmonics) |
| Policy architecture | MLP with hidden sizes $(32, 32)$ |
| Sampling integrator | Heun (RK2), 30 steps |
| Optimizer | Adam (Kingma, 2014) for both the actor and critic |

## K. Experimental Details of Sexually Transmitted Infection Testing

### K.1. Environment Details

We are given an undirected graph $\mathcal{G} = (\mathcal{V}, \mathcal{E})$ with $|\mathcal{V}| = n$ nodes. Each node $v \in \mathcal{V}$ has an unknown binary label $Y_v \in \{0, 1\}$, where $Y_v = 1$ indicates a *positive* individual and $Y_v = 0$ indicates a *negative* individual. Let $Y = (Y_v)_{v \in \mathcal{V}} \in \{0, 1\}^{\mathcal{V}}$ denote the vector of all labels. The labels are jointly distributed according to a prior distribution $\mathcal{P}$ over $\{0, 1\}^{\mathcal{V}}$ that is Markov with respect to $\mathcal{G}$ (e.g., specified by a graphical model). At the beginning of each episode, a realization $Y \sim \mathcal{P}$ is drawn and remains fixed throughout the episode.

Testing a node $v$ reveals its true label $Y_v$ and yields an immediate reward $r(Y_v)$, where $r : \{0, 1\} \to \mathbb{R}_{\geq 0}$ is a fixed reward function (for example, $r(1) = 1$ and $r(0) = 0$ for counting detected positives). The decision maker does not observe $Y$ directly and must decide which nodes to test adaptively based on past outcomes.

**State representation.** We encode the observable status of each node by a variable $X_t(v) \in \{-1, 0, 1\}$ at time $t$, where

$$X_t(v) = \begin{cases} -1, & \text{if node } v \text{ has not been tested yet (label unknown)}, \\ 0, & \text{if node } v \text{ has been tested and found negative}, \\ 1, & \text{if node } v \text{ has been tested and found positive}. \end{cases}$$

Let $X_t = (X_t(v))_{v \in \mathcal{V}} \in \{-1, 0, 1\}^{\mathcal{V}}$ denote the status vector at time $t$. The *MDP state* at time $t$ is

$$s_t = X_t.$$

The graph $\mathcal{G}$ and prior $\mathcal{P}$ are fixed and known, so they are treated as part of the problem instance rather than the dynamic state. Initially, no node has been tested, so $X_0(v) = -1$ for all $v \in \mathcal{V}$.

*Table 3.* Graph statistics for each disease network.

| Disease | Nodes | Edges | Positive (%) | Components |
|---------|-------|-------|--------------|------------|
| Chlamydia | 100 | 63 | 44.0 | 37 |
| Gonorrhea | 100 | 68 | 9.0 | 32 |
| HIV | 100 | 70 | 27.0 | 37 |
| Syphilis | 101 | 102 | 18.8 | 27 |

**Frontier and actions.** Let $\mathcal{C}_1, \ldots, \mathcal{C}_m$ be the connected components of $\mathcal{G}$, and fix a designated root node $\rho_j \in \mathcal{C}_j$ for each component according to a given priority rule (e.g., the node with highest marginal probability of being positive under $\mathcal{P}$).

Given a state $X_t$, we define the set of tested nodes $S_t = \{v \in \mathcal{V} : X_t(v) \in \{0, 1\}\}$. The *frontier* $\mathcal{F}(X_t)$ consists of nodes that are currently eligible for testing:

$$\mathcal{F}(X_t) = \{\rho_j : S_t \cap \mathcal{C}_j = \emptyset\} \cup \{v \in \mathcal{V} : X_t(v) = -1 \text{ and } \exists\, u \in S_t \text{ with } (u, v) \in \mathcal{E}\}.$$

Intuitively, in each connected component where no node has been tested, only the root is available; once a node in a component has been tested, all untested neighbors of tested nodes become available.

We consider a batched setting with a per-round testing budget $B \in \{1, \ldots, n\}$. At time $t$, the agent chooses an *action* $A_t \subseteq \mathcal{F}(X_t)$ satisfying $|A_t| \leq B$, representing the batch of nodes to be tested in parallel at time $t$.

**Definition K.1** (Batched Adaptive Frontier Exploration on Graphs (B-AFEG)). An instance of B-AFEG is specified by a quadruple $(\mathcal{G}, \mathcal{P}, \gamma, B)$, where $\mathcal{G}$ is the graph, $\mathcal{P}$ is the prior over true labels $Y$, $\gamma \in (0, 1]$ is a discount factor, and $B$ is the per-round testing budget.

At each time $t = 0, 1, 2, \ldots$:

1. The state is $s_t = X_t \in \{-1, 0, 1\}^{\mathcal{V}}$.

2. The agent chooses an action $A_t \subseteq \mathcal{F}(X_t)$ with $|A_t| \leq B$.

3. For each $v \in A_t$, the true label $Y_v$ is revealed and the reward contributed by $v$ is $r(Y_v)$. The immediate reward at time $t$ is
$$R_t = \sum_{v \in A_t} r(Y_v).$$

4. The state is updated deterministically by setting
$$X_{t+1}(v) = \begin{cases} Y_v, & \text{if } v \in A_t, \\ X_t(v), & \text{otherwise,} \end{cases}$$
for all $v \in \mathcal{V}$. Once $X_t(v) \in \{0, 1\}$, it remains fixed for the rest of the episode.

A policy $\pi$ maps each state $X_t$ to a distribution over feasible actions $A_t \subseteq \mathcal{F}(X_t)$ with $|A_t| \leq B$. The performance of a policy $\pi$ is measured by its expected discounted return

$$J(\pi) = \mathbb{E}\left[\sum_{t=0}^{\infty} \gamma^t R_t\right],$$

where the expectation is taken over $Y \sim \mathcal{P}$ and any randomness in $\pi$. The objective is to find an optimal policy

$$\pi^\star \in \arg\max_{\pi} J(\pi).$$

*Table 4.* Hyperparameters used for sexually transmitted infection testing.

| Hyperparameter | Setting |
|---|---|
| *Training schedule* | |
| Warmup steps | 1000 |
| Training episodes (`train_updates`) | 2000 |
| Batch size | 64 |
| *Core RL / optimization* | |
| Discount factor ($\gamma$) | 0.99 |
| Learning rate (Adam) | $4 \times 10^{-4}$ |
| Target update rate ($\tau$) | 0.005 |
| Delayed update | 2 |
| Reward scale | 1.0 |
| Gradient clipping (max norm) | 5.0 |
| *Policy training* | |
| Number of particles ($K$) | 12 |
| $\lambda$ schedule | start 2.0, end 0.8, steps 10000 (Syphilis: end 0.5, steps 30000) |
| Weight clip ($w_{\max}$) | 4.0 |
| vMF concentration ($\kappa$) | Chlamydia, HIV: 60.0 / Syphilis, Gonorrhea: 40.0 |
| vMF perturbations per candidate ($J$) | 1 |
| Flow steps | 36 |
| Q-norm clip | 3.0 |
| Q running beta | 0.05 |
| *Policy network (GIN)* | |
| Hidden dimension | 128 |
| GIN layers | 3 |
| Time embedding dim | 32 |
| Optimizer | Adam (for both the actor and critic) |

### K.2. Implementation Details

**Dataset.** The ICPSR dataset is publicly available.[5] For each disease, we construct a pruned contact network by iteratively sampling connected components from the full ICPSR graph until reaching a target size of roughly 100 nodes. To limit variability, we cap the overshoot at 10% beyond the target, since the final component added may exceed the threshold. We repeat this randomized construction multiple times and retain the subgraph with the largest number of positive nodes. Table 3 summarizes the resulting graph statistics and disease prevalence, and Fig. 6 visualizes the corresponding networks.

**Implementation.** At each iteration, the policy selects $B = 5$ nodes. For a fair comparison, all RL methods use the same critic-network architecture and the same number of warm-up steps, and are then trained until convergence. At test time, we report average return over 10 evaluation episodes. We use the hyperparameters in Table 4.

**Random baseline.** At each iteration, the policy selects up to $B$ frontier nodes uniformly at random for testing.

**Greedy baseline.** At each iteration, the policy selects the $B$ frontier nodes with the largest number of positively tested neighbors. This heuristic exploits network structure under the assumption that proximity to known positives increases the likelihood of finding additional infections.

## L. Computing Infrastructure

Experiments were conducted with 4× NVIDIA H200 GPUs (with 1.5 TB host memory and 64 CPU cores).

---

[5]`https://www.icpsr.umich.edu/web/ICPSR/studies/22140`

# M. Additional Experimental Results

**Scalability to larger problem sizes.** The main benchmarks in Section 5.1 use a fixed problem size ($N = 40$ patients, $B = 10$ budget). To assess how LSFLOW scales, we run Dynamic Scheduling at substantially larger sizes, denoted $N/B$ where $N$ is the number of patients and $B$ the budget; larger $N$ and $B$ make the combinatorial action space harder. We compare against SRL, the strongest baseline on this task. As shown in Table 5, LSFLOW continues to outperform SRL across all sizes, indicating that learning the flow policy on $\mathbb{S}^{m-1}$ remains effective as the action dimension grows.

*Table 5.* Reward on Dynamic Scheduling at increasing problem sizes ($N/B$). Higher is better; mean $\pm$ std over evaluation episodes.

| Method | 100/20 | 200/30 | 500/50 | 1000/80 |
|---|---|---|---|---|
| SRL | $39.62_{\pm 1.24}$ | $57.64_{\pm 1.44}$ | $111.02_{\pm 2.27}$ | $204.07_{\pm 1.49}$ |
| Ours | $\mathbf{47.72}_{\pm 1.86}$ | $\mathbf{69.27}_{\pm 1.85}$ | $\mathbf{131.58}_{\pm 1.53}$ | $\mathbf{218.77}_{\pm 2.97}$ |

**Robustness to the choice of solver.** Our framework treats the combinatorial solver as a black box, so the learned policy should not depend on a particular solver implementation. We verify this on Dynamic Intervention by swapping the solver backend among Gurobi, SCIP, and CPLEX while keeping everything else fixed. The resulting rewards are nearly identical (Gurobi: $17.21_{\pm 0.39}$; SCIP: $17.03_{\pm 0.59}$; CPLEX: $17.34_{\pm 0.47}$), confirming that LSFLOW is robust to the choice of solver.

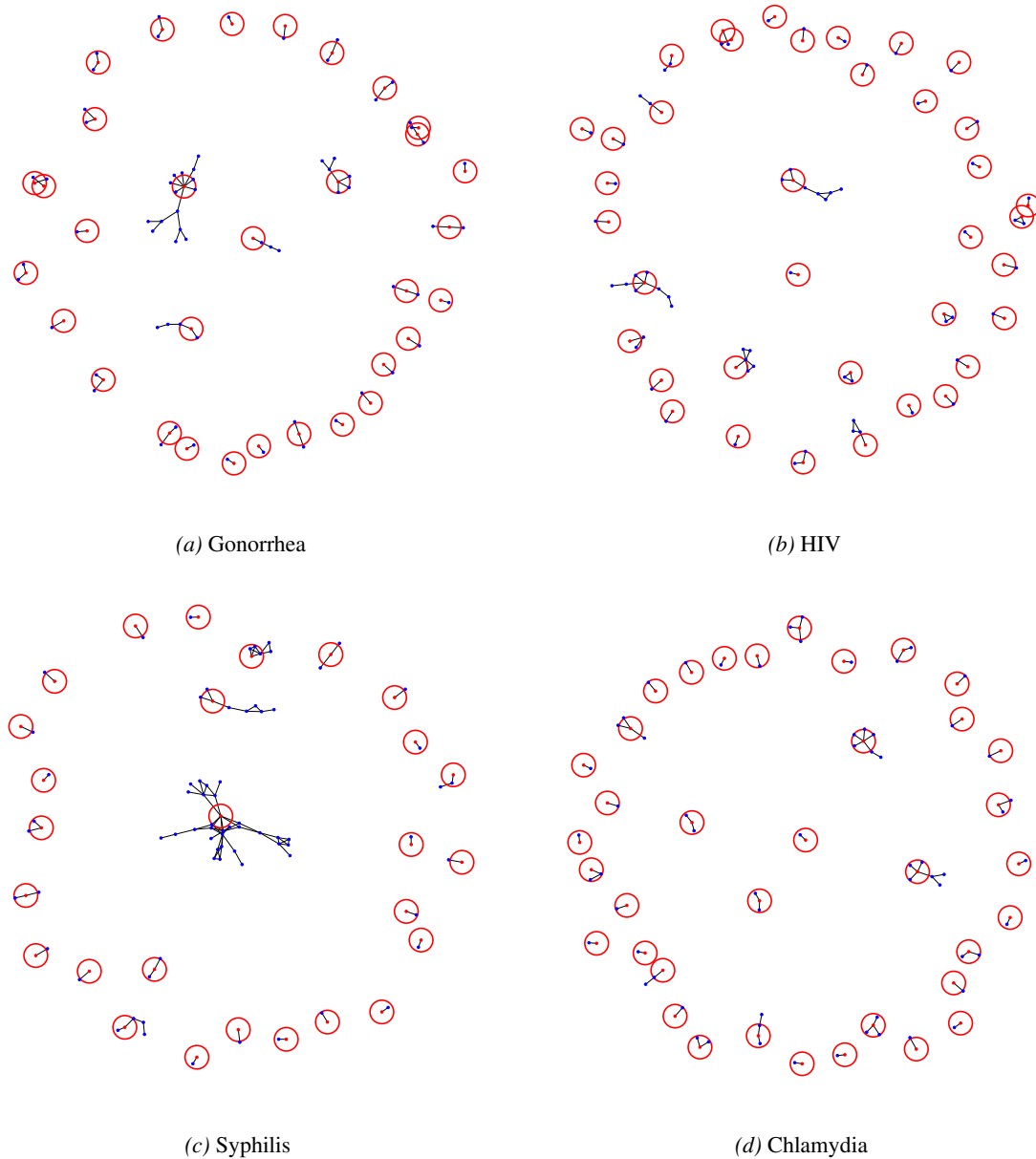

*(a)* Gonorrhea                    *(b)* HIV

*(c)* Syphilis                    *(d)* Chlamydia

*Figure 6.* Contact networks for each disease. Nodes represent individuals; edges represent reported sexual contacts. Frontier roots are circled in red.

