# OpenReview forum: "Latent Spherical Flow Policy for Reinforcement Learning with Combinatorial Actions"
_ICML.cc/2026/Conference — ICML 2026 spotlight_

### Official Review · Reviewer_AAjs · 2026-02-24

**Soundness:** 3
**Presentation:** 3
**Significance:** 3
**Originality:** 3
**Overall Recommendation:** 4
**Confidence:** 4

**Summary:**

This paper proposes LSFLOW, a method for RL with combinatorial action spaces that combines spherical flow matching with a combinatorial optimization (CO) solver. The key idea is a two-stage policy: (1) learn a stochastic distribution over cost-direction vectors on the unit sphere $\mathbb{S}^{m-1}$ via spherical flow matching, and (2) map each sampled cost vector to a feasible combinatorial action via an LP/MIP solver $a^\star(s,c) = \arg\min_{a \in \mathcal{A}(s)} c^\top a$. The paper introduces a latent-space critic to avoid repeated solver calls during policy updates, and a smoothed Bellman operator using von Mises–Fisher (vMF) kernels to handle the piecewise-constant value landscape induced by the solver mapping. Experiments on four combinatorial RL benchmarks and a real-world STI testing task show improvements over baselines.

**Compliance With Llm Reviewing Policy:**

Affirmed.

**Final Justification:**

The authors solved most of my questions in the rebuttal. However, some of them might be shown in the revision of this work. Now I would like to raise my score to 4, and check whether the answers are satisfying in the second phase.

**Key Questions For Authors:**

1.The Weighted spherical flow-matching objective in section 3.1 seems like an Energy-Weighted Flow Matching, which was proposed in (Energy-Weighted Flow Matching, Zhang et al, 2025). Maybe the authors should discuss more about that.

2.Can you provide a bound on $||Q_\kappa^\pi - Q^\pi ||^{\infty} $
as $\kappa \to \infty$? The empirical sensitivity to $\kappa$(Fig. 4) suggests this is important. Or at least we require $Q_\kappa^\pi\to Q^\pi$.

3.How does performance degrade as the action dimension $m$ increases? Have you tested on instances with larger $m$? Since high-dimension spheres might cause unexpected challenges for flow matching.

4. The paper uses Gurobi for all experiments. How sensitive is the method to solver quality or solver timeout? In practice, many CO solvers return approximate solutions. Does the theory (Theorem 3.5) still hold when $a^\star(s,c)$ is an approximate minimizer?

**Limitations:**

Yes

**Strengths And Weaknesses:**

Strength:
1. Lemma 3.1 formalizes that the solver output depends only on the direction of the cost vector. This justifies restricting the latent space to $ \mathbb{S}^{m-1} $, eliminating scale redundancy in a principled way. The spherical parameterization is conceptually clean and well motivated. Besides, Proposition 3.2 provides an expressivity guarantee, that any stochastic policy over $A(s)$ can be represented via an appropriate distribution over spherical cost vectors. This is a nontrivial result that clarifies the representational power of the framework.

2. The discontinuity of $c \mapsto a^\star(s,c)$ creates difficulties for value learning. The vMF-smoothed Bellman operator $\mathcal{T}{\kappa}^\pi $ resolves this by defining a smoothed fixed point $ Q_\kappa^\pi $. The contraction property and uniqueness of the fixed point are established, and the resulting value function is shown to be smooth. The analysis is natural, and the proofs appear correct.

3. Learning the critic in latent cost space avoids repeated solver calls during policy updates. The reported per-step speedup (approximately 360$\times$) is practically meaningful.

4. The reported average improvement over SRL is substantial. The inclusion of a real-world STI testing application strengthens the practical relevance.

5. Ablations on spherical geometry, smoothing strength $ \kappa $, and the latent critic clarify the contribution of each component.

Weakness:

1. The reweighted flow-matching objective (Eq. 4) uses exponentiated $Q$-values to weight samples. These method were discussed in prior works, so the authors may discuss about that. While the combinatorial setting and spherical parameterization are novel, the reweighting principle itself is not new and should be properly attributed.

2. The smoothed Bellman operator defines a fixed point $ Q_\kappa^\pi $, but the approximation error relative to the original value function $ Q^\pi $ is not analyzed. Besides, the experiment part suggests that $\kappa$'s selection influences the empirical results. I am wondering whether the bias take effect here.

3. All experiments use relatively small instances (e.g., $N=40$, $m \sim 100$). Flow matching on $ \mathbb{S}^{m-1} $ may become challenging as $m$ increases. I suggest the authors discussing a little about what may happen for a larger $m$.

4. The author acknowledge the limitation that the framework assumes a linear objective of the form, while many combinatorial problems involve nonlinear or quadratic objectives.

---

> ### Author Rebuttal · Authors · 2026-03-31
>
> We thank the reviewer for the insightful comments and respond to each point below.
>
> **W1 and Q1. The Weighted spherical flow-matching objective seems like an Energy-Weighted Flow Matching, which was proposed in Zhang et al, (2025).**
>
> Thank you for pointing out this related work. We want to clarify a few points:
>
> **Different setting.** Zhang et al. (2025) study offline RL, where reweighting is applied to a fixed dataset. Our method is in online combinatorial RL, where the reweighted objective supports iterative policy improvement through ongoing environment interaction.
>
> **Different derivation.** Our reweighting scheme follows the online diffusion policy literature, especially Ma et al. (2025a), which we already cite in Section 3.2. In particular, we connect exponential reweighting to KL-regularized policy improvement (Theorem H.1) following that line of work.
>
> We appreciate the suggestion and will add a discussion of Zhang et al. (2025) in the revised version.
>
> **W2 and Q2. Can we obtain $Q_{\kappa}^{\pi}\rightarrow Q^{\pi}$ ?**
>
> We thank the reviewer for this insightful question. We can establish the following pointwise convergence statement. Fix a reachable state $s$. For every direction $c \notin \mathcal B_s$, where $\mathcal B_s$ denotes the **solver-switching boundary**, i.e., the boundary between regions of cost directions that induce different feasible actions under the solver, we have
> $$
> Q_\kappa^\pi(s,c)\to Q^\pi(s,c)\qquad\text{as }\kappa\to\infty.
> $$
> Since $\mathcal B_s$ has surface measure zero, this implies $Q_\kappa^\pi(s,c)\to Q^\pi(s,c)$ for surface-almost-every $c\in\mathbb S^{m-1}$. Moreover, if $\pi(\cdot\mid s)$ is absolutely continuous with respect to surface measure, then the convergence also holds almost surely for $C\sim\pi(\cdot\mid s)$.
>
> The proof combines local constancy of the solver away from the switching boundary with concentration of the vMF kernel around its center. Under an additional uniform positive margin from the switching boundary assumption, one can further obtain an exponential convergence rate.
>
> **Connection to Fig. 4.** This explains the bias–stability tradeoff. Smaller $\kappa$ gives stronger smoothing, which stabilizes critic targets but introduces oversmoothing bias. Larger $\kappa$ reduces this bias, but the critic approaches the unsmoothed piecewise-constant value, making optimization noisier and less stable. The best-performing moderate $\kappa$ values in Fig. 4 reflect this tradeoff.
>
> We will add the proposition and discussion in the revised version.
>
> **W3 and Q3. I suggest the authors discussing a little about what may happen for a larger action dimension.**
>
> We thank the reviewer for this question. We ran additional experiments on Dynamic Scheduling with increasing problem size, where “100/20”, “200/30”, etc. denote $N/B$, with $N$ the number of patients and $B$ the number of workers. Larger $N$ and $B$ make the problem more challenging.
>
> We compare against the strongest baseline in this task, **SRL**, and obtain:
> - **100 / 20:** SRL = $39.62 \pm 1.24$, Ours = $47.72 \pm 1.86$
> - **200 / 30:** SRL = $57.64 \pm 1.44$, Ours = $69.27 \pm 1.85$
> - **500 / 50:** SRL = $111.02 \pm 2.27$, Ours = $131.58  \pm 1.53$
> - **1000 / 80:** SRL = $204.07 \pm 1.49$, Ours = $218.77 \pm  2.97$
>
> These results suggest that our method continues to outperform the strongest baseline as the problem size increases.
>
> **W4. The author acknowledge the limitation that the framework assumes a linear objective of the form, while many combinatorial problems involve nonlinear or quadratic objectives.**
>
> Please refer to our response to Q1 of Reviewer Kmrn for a detailed clarification: https://openreview.net/forum?id=07wwDFdi3k&noteId=iARHFU4ehS
>
> **Q4. The paper uses Gurobi for all experiments. ... Does the theory (Theorem 3.5) still hold when $a^{\star}(s,c)$ is an approximate minimizer?**
>
> We thank the reviewer for this question. For **Theorem 3.5**, exact inner optimality is not essential. If the exact solver $a^\star$ is replaced by a **feasible measurable approximate solver** $\hat a(s,c)\in\mathcal A(s)$, and the smoothed Bellman operator is redefined accordingly, the same contraction and smoothing argument still goes through. In particular, the modified operator remains a $\gamma$-contraction, admits a unique fixed point, and its fixed point is still $C^\infty$ in $c$. The only change is in interpretation: the fixed point now corresponds to the smoothed value function induced by the approximate solver $\hat a$, rather than the exact argmin solver.
>
> Empirically, we also tested different solver backends on **Dynamic Intervention** and obtained very similar results: **Gurobi** ($17.21 \pm 0.39$), **SCIP** ($17.03 \pm 0.59$), and **CPLEX** ($17.34 \pm 0.47$). This suggests that the method is reasonably robust to solver choice in practice.

---

> > ### Author Rebuttal · Reviewer_AAjs · 2026-04-02
> >
> > The author's rebuttal solves my question, so I will raise my score up to 4.

---

> > > ### Author Response · Authors · 2026-04-08
> > >
> > > We thank the reviewer for the insightful comments, which will help improve our manuscript. We will revise accordingly.

---

### Official Review · Reviewer_4kdz · 2026-03-12

**Soundness:** 3
**Presentation:** 3
**Significance:** 3
**Originality:** 3
**Overall Recommendation:** 5
**Confidence:** 4

**Summary:**

This paper investigates the application of flow matching to reinforcement learning (RL) problems with combinatorial action spaces. A core challenge in this setting is that the high dimensionality of combinatorial action spaces and their complex feasibility constraints render direct continuous parameterization of policies infeasible. To address this, the work parameterizes linear objectives of the combinatorial problem as a continuous latent space, and leverages a combinatorial optimization solver to map latent space outputs to feasible discrete actions. Given that the latent action’s efficacy depends solely on its directional characteristics, spherical flow matching is adopted to learn a highly expressive policy over this latent space. Additionally, the critic network is trained directly in the latent space to circumvent the heavy computational overhead of repeated solver calls, and a smoothed Bellman operator is proposed to mitigate discontinuities in value estimates induced by the discrete optimization solver. Experimental results demonstrate that the expressive flow-based policy learned via this framework yields substantial performance gains over state-of-the-art combinatorial RL methods.

**Compliance With Llm Reviewing Policy:**

Affirmed.

**Key Questions For Authors:**

1. Could the authors investigate and compare the performance of the proposed framework with direct Q-value maximization-based policy optimization?

**Limitations:**

yes

**Strengths And Weaknesses:**

# Strengths
1. *Originality*: The paper presents a novel solution that integrates modern generative flow-based models into combinatorial RL problems with rigid action space feasibility constraints. The design choices, including the spherical latent action space, and smoothed critic updates optimized directly in this latent space are both motivated and empirically effective.
2. *Soundness*: The proposed method outperforms existing baselines with strong convergence performance, while also achieving notable computational efficiency in terms of training time.
3. *Presentation*: The paper is well-structured and clearly written. All technical components and design decisions are thoroughly motivated with clear connections to the core challenges of combinatorial RL.
# Weaknesses
1.  The method relies on formulating the target combinatorial RL problem as a constrained optimization task with linear constraints exclusively. This restrictive formulation might limit the method’s applicability to other problems.
2. The adopted Q-weighted policy extraction strategy may compromise the final convergence performance of the method, which is studied in [1]. If we have a high-capacity policy network, this approach can cause the policy to assign non-trivial probability mass to suboptimal actions. Alternative policy optimization strategies, such as direct Q-value maximization for policy update [2], could be explored.

[1] https://arxiv.org/pdf/2406.09329

[2] https://arxiv.org/abs/2506.12811

---

> ### Author Rebuttal · Authors · 2026-03-31
>
> We thank the reviewer for the insightful comments and respond to each point below.
>
> **W1. The method relies on formulating the target combinatorial RL problem as a constrained optimization task with linear constraints exclusively. This restrictive formulation might limit the method’s applicability to other problems.**
>
>
>
> We would like to clarify a misunderstanding. The linear form $c^\top a$ appears only as the latent-space interface between the learned policy and the combinatorial solver; it does not require the underlying combinatorial RL problem to have either a linear objective or linear constraints.
>
>
> **First, our framework does not require the underlying combinatorial RL objective to be linear.** Proposition 3.2 shows that the solver-induced decoder $a^\star(s,c) = \arg\min_{a \in \mathcal{A}(s)} c^\top a$ is already expressive enough to represent **any stochastic policy over $\mathcal{A}(s)$** by learning an appropriate distribution over $c$. The linear form is therefore a decoder parameterization for feasible action generation, not a restriction on the problem class. This is further evidenced by our experiments, where LSFLOW achieves the best performance in nonlinear settings (e.g., the Dynamic Intervention task).
>
>
> **Second, our framework does not require the constraints defining $A(s)$ to be linear.** The feasible set $A(s)$ can encode much more general combinatorial structure, as long as there exists a solver that can optimize over it under the given parameterization. In practice, many off-the-shelf solvers support substantially richer constraint classes; for example, SCIP [1] and Bonmin [2] can handle mixed-integer nonlinear programs. We will revise the paper to make this point clearer.
>
> In fact, this linear formation in the latent space is one of the key innovations of our design, with two concrete advantages: (1) it enables efficient solver computation, since combinatorial programs with linear objectives are typically much easier to solve than nonlinear or quadratic ones; and (2) it justifies the spherical parameterization via Lemma 3.1 (positive scale invariance), which in turn enables spherical flow matching and the compact latent representation underlying our method.
>
> [1] https://www.scipopt.org/
> [2] https://www.coin-or.org/Bonmin/
>
>
> **W2 and Q1. The adopted Q-weighted policy extraction strategy may compromise the final convergence performance of the method, which is studied in [1]. If we have a high-capacity policy network, this approach can cause the policy to assign non-trivial probability mass to suboptimal actions. Alternative policy optimization strategies, such as direct Q-value maximization for policy update [2], could be explored.**
>
>
>
> We thank the reviewer for this suggestion and agree that more direct value-aware policy optimization is an interesting direction.
>
> Regarding [1], the evidence is empirical and obtained in standard offline RL benchmarks rather than online combinatorial RL. Its conclusions therefore do not directly transfer to our setting.
>
> Regarding [2], direct application in our combinatorial RL setting faces two concrete obstacles. First, applying [2] in action space would require end-to-end optimization through the solver-induced action map, precisely the expensive and non-smooth route our method avoids, and would reintroduce repeated solver calls into the policy optimization loop. Second, adapting [2] to latent cost space is non-trivial for two reasons: the regularization in [2] is derived for Euclidean flow policies and would require a new sphere-aware derivation in our setting; and the solver map is many-to-one, so a high-value feasible action corresponds to an entire region of latent cost directions. Therefore, adapting [2] would require an additional design choice for constructing the latent reference distribution.
>
> We also conduct a preliminary experiment on the Dynamic Scheduling task, directly adding a value maximization term to our weighted spherical flow matching loss. However, performance drops from **28.85** to **21.58**. We hypothesize that this is because direct backpropagation through the flow sampler destabilizes RL training in the combinatorial setting. Developing a stable direct value maximization approach for combinatorial RL is therefore an interesting but non-trivial direction for future work, and we will include this discussion in the revision.

---

> > ### Author Rebuttal · Reviewer_4kdz · 2026-04-03
> >
> > I appreciate the authors' efforts in the response. My concerns are fully resolved. Since the original score is already 5, I will keep it as is.

---

> > > ### Author Response · Authors · 2026-04-08
> > >
> > > We thank the reviewer for the insightful comments, which will help improve our manuscript. We will revise accordingly.

---

### Official Review · Reviewer_g7rD · 2026-03-13

**Soundness:** 3
**Presentation:** 3
**Significance:** 3
**Originality:** 3
**Overall Recommendation:** 5
**Confidence:** 4

**Summary:**

The paper proposes LSFLOW, a reinforcement learning algorithm designed for environments with combinatorial action spaces. To enforce hard feasibility constraints while maintaining policy expressiveness, the authors decompose the decision-making process into two stages: learning a continuous stochastic policy over a latent cost space using spherical flow matching, and mapping the sampled latent cost vectors to discrete valid actions via a deterministic combinatorial optimization (CO) solver. To address the computational bottleneck of repeatedly invoking the CO solver during policy optimization, the authors train the critic directly in the latent cost space. Furthermore, to resolve the discontinuous value landscape induced by the solver's piecewise-constant mapping, a von Mises-Fisher (vMF) smoothed Bellman operator is introduced. The method is evaluated on four public combinatorial RL benchmarks and a graph-based STI testing task, achieving state-of-the-art performance.

**Compliance With Llm Reviewing Policy:**

Affirmed.

**Final Justification:**

My final recommendation is Accept. The primary strength of this paper lies in its novelty and significance. By mapping the expressiveness of spherical flow matching into a structured discrete space via a combinatorial optimization solver, LSFLOW successfully bridges the gap between continuous generative policies and strict discrete feasibility constraints. Empirically, the algorithm demonstrates exceptionally strong performance. It outperforms current state-of-the-art baselines (including SEQUOIA and SRL) by a substantial margin of 20.6% on average across benchmark tasks.

Regarding my initial reservation about the reliance on the solver's cold-start search and potential inference delays in extremely constrained environments, the authors' rebuttal effectively resolved my concerns. The provided latency metrics confirm LSFLOW remains practically competitive. More importantly, the authors correctly contextualized the cold-start limitation as an open problem across the entire combinatorial RL community.

In light of the significant empirical gains, the algorithmic novelty, and the well-argued rebuttal, I have raised my score to Accept.

**Key Questions For Authors:**

1. Unlike action-space methods that can provide structural warm-starts, LSFLOW requires the solver to find a feasible solution purely from a cost vector $c$. Have you observed scenarios (e.g., highly constrained environments) where the solver struggles to find a feasible action efficiently due to the lack of structural initialization from the policy?
2. Could you provide concrete wall-clock inference times (e.g., latency per step in milliseconds) for the tested environments?

**Limitations:**

Yes.

**Strengths And Weaknesses:**

## Strengths
1. **Clear Motivation and Sound Methodological Design**: The architecture is well-reasoned, with each component addressing a specific, identifiable challenge in combinatorial RL:
    - *Spherical Flow Matching*: Recognizing that the downstream CO solver is positively scale-invariant (Lemma 3.1), the authors restrict the latent space to the unit sphere $\mathbb{S}^{m-1}$. This avoids wasting network capacity on learning magnitude representations.
    - *Latent-Space Critic*: By moving the critic to the continuous latent space, the method amortizes the computational cost of the CO solver, bypassing the need to differentiate through or repeatedly call the solver within the inner loop of policy optimization.
    - *vMF Smoothed Bellman Operator*: The introduction of the vMF kernel effectively mitigates the gradient variance issues caused by the solver's step-function-like behavior. Theorem 3.5 provides rigorous guarantees that this operator yields a unique fixed point and a $C^\infty$ smoothed value function.
2. **Comprehensive Ablation Studies**: The authors provide targeted ablations that isolate and validate the necessity of their design choices. The comparison between spherical flow and unconstrained Euclidean flow confirms the practical benefit of embedding the policy on the sphere. The sensitivity analysis on the vMF concentration parameter $\kappa$ demonstrates that moderate smoothing is required, correctly identifying the trade-off between target stability (variance reduction) and signal preservation (bias). The evaluation of the latent-space critic against an action-space baseline quantifies the computational necessity of the proposed architecture. It demonstrates a substantial reduction in wall-clock time per policy update (from approximately 6 minutes to under 1 second) by avoiding solver calls during the policy optimization loop.
3. **Strong Empirical Performance**: The algorithm empirically outperforms current SOTA baselines (including SEQUOIA and SRL) by an average of 20.6% on the benchmark tasks. Additionally, it shows a clear advantage in training efficiency (e.g., ~3.0x faster than SRL) because it does not require solver-in-the-loop differentiation. The application to the STI testing network further illustrates the method's compatibility with graph neural networks (GIN), a flexibility not supported by prior MIP-embedded methods.
4. **Novelty and Significance**: The methodology presents a novel intersection of continuous generative modeling and combinatorial optimization. By mapping the expressiveness of flow matching into a structured discrete space via a solver, the authors provide a principled mechanism to handle combinatorial action spaces without compromising the structural flexibility of the policy or the critic. This framework offers a scalable and theoretically sound template for future research in constrained sequential decision-making.

## Weaknesses
1. **Over-reliance on the Solver for Feasibility Discovery**: The proposed latent-space optimization completely shifts the burden of finding a feasible solution $\arg\min_{a \in \mathcal{A}(s)} c^\top a$ onto the CO solver. In standard action-space optimization methods, even if a neural network outputs an action $a_{raw}$ that violates constraints, $a_{raw}$ often serves as a near-feasible "warm-start" or structural heuristic, making it relatively easy for a projection or a local search algorithm to snap it to a valid solution. In contrast, LSFLOW generates a latent cost vector $c \in \mathbb{S}^{m-1}$. The solver receives a linear objective but no structural initialization from the policy. In highly constrained environments where finding any feasible solution is computationally hard (e.g., sparse feasible regions or tight routing budgets), this complete reliance on the solver's cold-start search capability could result in severe performance degradation or prohibitive latency.
2. **Inference Latency**: Although training efficiency is improved via the latent critic, action execution at inference time still requires solving a combinatorial optimization problem at every step. A brief discussion on the practical latency limits of this approach for real-time control would be beneficial.

---

> ### Author Rebuttal · Authors · 2026-03-31
>
> We thank the reviewer for the insightful comments and respond to each point below.
>
> **W1 and Q1.  Unlike action-space methods that can provide structural warm-starts, LSFLOW requires the solver to find a feasible solution purely from a cost vector $c$. Have you observed scenarios (e.g., highly constrained environments) where the solver struggles to find a feasible action efficiently due to the lack of structural initialization from the policy?**
>
> We thank the reviewer for this thoughtful point. A key reason we adopt this **linear objective interface** is that it provides a practically attractive balance between tractability and expressiveness. On the one hand, combinatorial programs with linear objectives are typically much easier to solve than nonlinear or quadratic ones. On the other hand, Proposition 3.2 shows that, for a finite feasible action set, the solver-induced decoder $a^\star(s,c)=\arg\min_{a\in A(s)} c^\top a$ is already expressive enough to represent **any stochastic policy** over $A(s)$. In this sense, the linear objective is a deliberate design choice that preserves expressiveness while keeping the solver interface relatively simple.
>
> We agree that our method does not provide a primal warm-start in the same sense as methods that output a near-feasible structured action. However, the warm-start advantage of action-space methods is **problem-dependent rather than universal**. They are useful only when the raw infeasible action contains structural information that can be exploited efficiently by a downstream projection or repair procedure. This need not always be the case in combinatorial settings, since feasibility may depend on globally coupled constraints and repairing an infeasible structured output may still be nontrivial. For example, in classical constrained problems such as Vehicle Routing Problem with Time Windows (VRPTW), even finding a feasible solution can already be computationally hard for a fixed number of vehicles [1], so one cannot generally assume that repairing an infeasible structured output will be cheap.
>
>
> We would further note that this issue is not unique to our method. Strongest baselines such as **SRL** and **SEQUOIA** also rely on downstream combinatorial optimization and do not naturally avoid the underlying feasibility burden. Empirically, we did **not** observe the lack of primal warm-starts to be the dominant bottleneck on the public benchmark tasks studied in the paper (Table 1). These tasks already involve nontrivial combinatorial feasibility constraints. In these settings, LSFLOW is about $3\times$ faster than SRL in average training time and much faster than SEQUOIA, precisely because we avoid repeated solver calls inside policy optimization and require only a single solver call per environment step.
>
> That said, we agree that explicit solver warm-starting is an interesting and valuable future direction, especially for extremely hard instances. For example, one could imagine augmenting our framework with a learned primal heuristic or solver-aware initialization mechanism. We will add this discussion in the revised version.
>
> [1] https://backend.orbit.dtu.dk/ws/files/5065984/BK_PhD_thesis_final_060110.pdf?utm_source=chatgpt.com
>
> **W2 and Q2. Inference Latency: Although training efficiency is improved via the latent critic, action execution at inference time still requires solving a combinatorial optimization problem at every step. A brief discussion on the practical latency limits of this approach for real-time control would be beneficial.**
>
>
> We thank the reviewer for this helpful suggestion. To assess practical inference latency, we additionally compare the average inference time across the four tasks in Section 5.1 against the two main combinatorial DRL baselines, SRL and SEQUOIA. The average inference times are: SRL: 40.23s, SEQUOIA: 84.98s, and Ours: 47.42s.
>
> These results suggest that our method remains practically competitive at inference time among combinatorial RL approaches. Compared with SRL, our method is only slightly  slower, which is expected since both methods rely on solving a combinatorial optimization problem at each step, and our method additionally includes a flow-sampling step. Compared with SEQUOIA, our method is substantially faster, since SEQUOIA solves a combinatorial optimization problem whose complexity grows with the size of the embedded value network.
>
> For real-time deployment, there is still room for further improvement. First, as discussed above, incorporating solver warm-starting could further reduce the combinatorial optimization overhead. Second, the extra cost from the flow component may also be reduced, since there is a growing body of work on accelerated sampling for flow-based generative models. We will add this discussion to the revised version.

---

> > ### Author Rebuttal · Reviewer_g7rD · 2026-04-01
> >
> > I have read the authors' rebuttal and appreciate the concrete inference latency metrics provided.
> >
> > While the latency numbers on the benchmark tasks (Ours: 47.42s vs SEQUOIA: 84.98s) are competitive, I maintain a strong reservation regarding the method's scalability to environments with extremely tight combinatorial constraints. The provided benchmarks may not fully expose the limitations of relying entirely on a solver's cold-start search. In problem instances where finding *any* initial feasible solution is intrinsically hard (e.g., highly constrained VRPTW), the absence of a structural primal warm-start from the policy could still lead to prohibitive inference delays. The empirical efficiency demonstrated here likely indicates that the feasible regions of the chosen tasks are relatively benign for the underlying solver.
> >
> > Furthermore, the authors' suggestion of "augmenting our framework with a learned primal heuristic or solver-aware initialization mechanism" is acknowledged, but this represents a non-trivial algorithmic challenge rather than a straightforward extension. Mapping a continuous latent cost vector to a structured, near-feasible primal initialization effectively requires solving the exact representation problem that LSFLOW circumvents by delegating to the CO solver.
> >
> > Given these considerations, my concerns are partially resolved. I agree the proposed latent spherical flow performs well within the scope of the tested benchmarks. Therefore, I will maintain my score of Weak Accept.

---

> > > ### Author Response · Authors · 2026-04-02
> > >
> > > **While the latency numbers on the benchmark tasks (Ours: 47.42s vs SEQUOIA: 84.98s) are competitive, I maintain a strong reservation regarding the method's scalability to environments with extremely tight combinatorial constraints. The provided benchmarks may not fully expose the limitations of relying entirely on a solver's cold-start search. In problem instances where finding any initial feasible solution is intrinsically hard (e.g., highly constrained VRPTW), the absence of a structural primal
> > > warm-start from the policy could still lead to prohibitive inference delays. The empirical efficiency demonstrated here likely indicates that the feasible regions of the chosen tasks are relatively benign for the underlying solver.**
> > >
> > > We thank the reviewer for this helpful comment. We agree that extreme constrained settings may cause delays, and we will acknowledge this more explicitly in the revised paper. However, we would also like to clarify a few points.
> > >
> > > First, this challenge is not unique to our method. SOTA baselines such as SRL and SEQUOIA also rely on cold-start solver search at inference time. To the best of our knowledge, there is currently no general action-space RL method that is effective across diverse combinatorial RL settings. We therefore view this as an open challenge for this domain, rather than a limitation specific to LSFLOW. We thank the reviewer for highlighting this point, and we will include a clearer discussion in the revised version.
> > >
> > > Second, even when a method outputs an infeasible structured candidate, this does not necessarily translate into faster search after projection, repair, or local search. This advantage is problem-dependent rather than universal: in many combinatorial problems, feasibility is governed by globally coupled constraints, so an infeasible structured output is not necessarily close to any feasible solution, and repairing it may still be nontrivial.
> > >
> > > **Furthermore, the authors' suggestion of "augmenting our framework with a learned primal heuristic or solver-aware initialization mechanism" is acknowledged, but this represents a non-trivial algorithmic challenge rather than a straightforward extension. Mapping a continuous latent cost vector to a structured, near-feasible primal initialization effectively requires solving the exact representation problem that LSFLOW circumvents by delegating to the CO solver.**
> > >
> > >
> > > We agree that augmenting LSFLOW with a learned warm-start is non-trivial, and we will revise our wording accordingly. At the same time, we would clarify that useful solver guidance need not take the form of a structured near-feasible primal initialization. In the learning-for-optimization literature, learned components have **successfully accelerated** solver search through other forms of guidance, such as branching policies (Elias et al., 2016, Gasse et al., 2019) and instance-aware separator configuration (Li et al., 2023), without requiring the model to directly generate a structured near-feasible primal solution. In this sense, we view solver-aware guidance as a plausible auxiliary acceleration layer rather than as solving the exact structured representation problem that LSFLOW intentionally avoids. We thank the reviewer again for this helpful discussion and will revise the paper to better clarify both this limitation and this future direction!
> > >
> > >
> > > [1] Khalil, Elias, et al. "Learning to branch in mixed integer programming." Proceedings of the AAAI conference on artificial intelligence. Vol. 30. No. 1. 2016.
> > >
> > > [2] Gasse, Maxime, et al. "Exact combinatorial optimization with graph convolutional neural networks." Advances in neural information processing systems 32 (2019).
> > >
> > > [3] Li, Sirui, et al. "Learning to configure separators in branch-and-cut." Advances in Neural Information Processing Systems 36 (2023): 60021-60034.

---

### Official Review · Reviewer_Kmrn · 2026-03-13

**Soundness:** 3
**Presentation:** 3
**Significance:** 3
**Originality:** 3
**Overall Recommendation:** 5
**Confidence:** 3

**Summary:**

The paper focuses on reinforcement learning problems with combinatorial action spaces. To mitigate the challenges, the paper proposes the use of a solver-induced spherical flow policy. To avoid a computational bottleneck, latent space training is used for both the critic and actor networks. The method is compared against relevant baselines and shows performance improvements in the selected domains.

**Compliance With Llm Reviewing Policy:**

Affirmed.

**Final Justification:**

The authors have addressed my concerns in their rebuttal on the soundness of the results and clarified the linear solver in terms of it's expressiveness. I therefore have raised the recommendation to an accept.

**Key Questions For Authors:**

- Are there potential limitations from the solver-induced flow policy objective being a linear objective?
- Beyond the domains used in evaluations, can the authors give examples of the type of domains where the current method is applicable?
- In the paper it reads 'For a fair comparison, all RL methods use the same critic-network architecture', however, one architecture may benefit the proposed method more than the related baselines (also true for shared hyperparameters) - how much does the architecture differ from the baseline architectures used in their corresponding paper(s)?

**Limitations:**

yes

**Strengths And Weaknesses:**

Strengths:
- The use of spherical flows for combinatorial action challenges is a novel direction.
- The performance of the method against relevant baselines showcases the methods work well in certain domain settings.
- The paper provides relevant theoretical details of the proposed approach.

Neutral:
- The comparative evaluation between the proposed approach and SRL in the STI testing domain does not show any clear improvement in half of the domains (HIV and Syphilis).

Weaknesses:
- The paper is generally well written, however, providing pseudocode would have made the approach more readable.

---

> ### Author Rebuttal · Authors · 2026-03-31
>
> We thank the reviewer for the insightful comments and respond to each point below.
>
> **W1. The paper is generally well written, however, providing pseudocode would have made the approach more readable.**
>
>
> The pseudocode is listed in Appendix B, specifically Algorithm 1 on page 13, and referenced in the main text (section 3.4).
>
> **Q1. Are there potential limitations from the solver-induced flow policy objective being a linear objective?**
>
> We would like to clarify that the linear objective $c^\top a$ appears only as the **latent-space interface** between the learned policy and the combinatorial solver; it does not restrict the class of combinatorial RL problems the framework can handle. Proposition 3.2 shows that the solver-induced decoder $a^\star(s,c) = \arg\min_{a \in \mathcal{A}(s)} c^\top a$ is already expressive enough to represent **any stochastic policy over $\mathcal{A}(s)$** by learning an appropriate distribution over $c$. The linear form is therefore a decoder parameterization for feasible action generation, not a restriction on the problem class. This is further evidenced by our experiments, where LSFLOW achieves the best performance in nonlinear settings (e.g., the Dynamic Intervention task).
>
> In fact, this linear formation in the latent space is one of the key innovations of our design, with two concrete advantages: (1) it enables efficient solver computation, since combinatorial programs with linear objectives are typically much easier to solve than nonlinear or quadratic ones; and (2) it justifies the spherical parameterization via Lemma 3.1 (positive scale invariance), which in turn enables spherical flow matching and the compact latent representation underlying our method.
>
> We note that future work may extend the current latent space linear-solver formulation to richer settings, such as multi-objective or robust formulations. These formulations would extend the scope, but our current method already enables expressiveness of a wide class of objectives, including nonlinear or quadratic.
>
>
> **Q2. Beyond the domains used in evaluations, can the authors give examples of the type of domains where the current method is applicable?**
>
> Our method is applicable to general sequential decision-making problems with combinatorial action spaces and hard feasibility constraints. Beyond our benchmarks, natural examples include:
>
> 1. **Power systems:** sequential unit commitment where the action is a subset of generators to activate at each timestep, subject to operational constraints such as capacity and ramping limits.
>
> 2. **Online advertising:** repeated budget allocation where the action is a subset of ad slots to bid on at each timestep, subject to campaign budget and pacing constraints.
>
> 3. **Ride-sharing and mobility:** repeated driver-passenger matching where at each step the agent selects a feasible set of matches, subject to capacity and geographic constraints.
>
> **Q3. In the paper it reads 'For a fair comparison, all RL methods use the same critic-network architecture', however, one architecture may benefit the proposed method more than the related baselines (also true for shared hyperparameters) - how much does the architecture differ from the baseline architectures used in their corresponding paper(s)?**
>
> We follow the critic architecture from the original benchmark [1] (3-layer MLP) without any tuning for our method, and adopt all shared hyperparameters (learning rate, batch size, discount factor, etc.) directly from the same source. We therefore believe this setup produces a fair comparison.
>
> [1] https://github.com/lily-x/combinatorial-rmab

---

> > ### Author Rebuttal · Reviewer_Kmrn · 2026-04-03
> >
> > I thank the authors for their rebuttal. All of my concerns have been addressed. I will raise my score from 4 to 5 (Accept).

---

> > > ### Author Response · Authors · 2026-04-08
> > >
> > > We thank the reviewer for the insightful comments, which will help improve our manuscript. We will revise accordingly.

---

### Decision · Program_Chairs · 2026-04-30

**Decision:**

Accept (spotlight)

**Comment:**

This paper proposes LSFlow, a latent spherical flow policy for reinforcement learning with combinatorial action spaces, combining a stochastic latent-space policy with a combinatorial solver to guarantee feasibility by design. Reviewers found the paper technically strong and well motivated, highlighting in particular the novelty of bringing flow-based generative policies into combinatorial RL, the principled latent-space formulation, and the smoothed Bellman operator used to stabilize value learning under solver-induced discontinuities.

The empirical evaluation was also viewed positively: the method consistently outperforms strong baselines across a diverse set of combinatorial RL benchmarks, and the ablations were seen as thorough and informative in validating the main design choices. The main concerns raised during discussion focused on practical limitations rather than flaws in the core contribution, especially the dependence on a downstream solver for feasible action generation and potential inference-time latency in highly constrained settings. However, the rebuttal addressed these concerns constructively by clarifying the role and expressiveness of the linear solver interface, providing concrete inference latency comparisons, and appropriately positioning extreme constraint regimes as an open challenge for the broader area rather than a problem unique to this method. One reviewer explicitly raised their score to accept after rebuttal (Kmrn), while the other maintained a positive assessment with only a limited reservation about especially difficult constraint regimes (g7rD). Overall, the paper presents a novel, technically sound, and empirically strong contribution to combinatorial reinforcement learning.